# Diffusion Models are Certifiably Robust Classifiers

**Huanran Chen**[1,2]**, Yinpeng Dong**[1,2]**, Shitong Shao**[1]**, Zhongkai Hao**[1]**,**
**Xiao Yang**[1]**, Hang Su**[1,3]**, Jun Zhu**[1,2*]

[1]Dept. of Comp. Sci. and Tech., Institute for AI, Tsinghua-Bosch Joint ML Center, THBI Lab
BNRist Center, Tsinghua University, Beijing, 100084, China   [2]RealAI
[3] Zhongguancun Laboratory, Beijing, China
huanran.chen@outlook.com  {dongyinpeng, dcszj}@mail.tsinghua.edu.cn

## Abstract

Generative learning, recognized for its effective modeling of data distributions, offers inherent advantages in handling out-of-distribution instances, especially for enhancing robustness to adversarial attacks. Among these, diffusion classifiers, utilizing powerful diffusion models, have demonstrated superior empirical robustness. However, a comprehensive theoretical understanding of their robustness is still lacking, raising concerns about their vulnerability to stronger future attacks. In this study, we prove that diffusion classifiers possess $O(1)$ Lipschitzness, and establish their certified robustness, demonstrating their inherent resilience. To achieve non-constant Lipschitzness, thereby obtaining much tighter certified robustness, we generalize diffusion classifiers to classify Gaussian-corrupted data. This involves deriving the evidence lower bounds (ELBOs) for these distributions, approximating the likelihood using the ELBO, and calculating classification probabilities via Bayes' theorem. Experimental results show the superior certified robustness of these Noised Diffusion Classifiers (NDCs). Notably, we achieve over 80% and 70% certified robustness on CIFAR-10 under adversarial perturbations with $\ell_2$ norms less than 0.25 and 0.5, respectively, using a single off-the-shelf diffusion model without any additional data.

## 1   Introduction

Despite the unprecedented success of discriminative learning [32, 22], they are vulnerable to adversarial examples, which are generated by imposing human-imperceptible perturbations on natural examples but can mislead target models into making erroneous predictions [44, 8]. To improve the robustness of discriminative learning, numerous defense techniques have been developed [29, 51, 35, 27, 34]. However, since discriminative models are directly trained for specific tasks, they often find shortcuts in the objective function, exhibiting non-robust nature [7, 36]. For example, adversarial training exhibits poor generalization against unseen threat models [45, 34], and purification-based methods typically cannot completely remove adversarial perturbations, leaving subsequent discriminative classifiers still affected by these perturbations [1, 25, 3, 15].

On the contrary, generative learning is tasked with modeling the entire data distribution, which offers a degree of inherent robustness without any adversarial training [53, 9]. As the current state-of-the-art generative approach, diffusion models provide a more accurate estimation of the score function across the entire data space. Thus, they have been effectively utilized as generative classifiers for robust classification, known as diffusion classifiers [26, 3, 5]. Specifically, they calculate the classification probability $p(y|\mathbf{x}) \propto p(\mathbf{x}|y)p(y)$ through Bayes' theorem and approximate the log likelihood $\log p(\mathbf{x}|y)$ via the evidence lower bound (ELBO). This method establishes a connection between robust classification and the fast-growing field of pre-trained generative models. Although

---

*The corresponding Author.

Code is available at `https://github.com/huanranchen/NoisedDiffusionClassifiers`.

38th Conference on Neural Information Processing Systems (NeurIPS 2024).

promising, there is still a lack of rigorous theoretical analysis, raising questions about whether their robustness is overestimated and whether they will be vulnerable to (potentially) stronger future adaptive attacks. In this work, we use theoretical tools to derive the certified robustness of diffusion classifiers, fundamentally address these concerns, and gain a deeper understanding of their robustness.

We begin by analyzing the smoothness of diffusion classifiers through the derivation of their Lipschitzness. We prove that diffusion classifiers possess an $O(1)$ Lipschitz constant, demonstrating their inherent robustness. This allows us to certify the robust radius of diffusion classifiers by dividing the gap between predictions on the correct class and the incorrect class by their Lipschitz constant. Although we obtain a non-trivial certified radius, it could be much tighter if we could derive non-constant Lipschitzness (i.e., the Lipschitzness at each point). Randomized smoothing [6, 38], a well-researched technique, allows us to obtain tighter Lipschitzness based on the output at each point. However, randomized smoothing requires the base classifier (e.g., diffusion classifiers) to process Gaussian-corrupted data $\mathbf{x}_\tau$, where $\tau$ is the noise level. To address this, we generalize diffusion classifiers to calculate $p(y|\mathbf{x}_\tau)$ by estimating $\log p(\mathbf{x}_\tau|y)$ using its ELBO and then calculating $p(y|\mathbf{x}_\tau)$ using Bayes' theorem. We named these generalized diffusion classifiers as **Noised Diffusion Classifiers**. Hence, the core problem becomes deriving the ELBO for noisy data.

Naturally, we conceive to generalize the ELBO in Sohl-Dickstein et al. [41] and Kingma et al. [17] to $\tau \neq 0$, naming the corresponding diffusion classifier the Exact Posterior Noised Diffusion Classifier (EPNDC). EPNDC achieves state-of-the-art certified robustness among methods that do not use extra data. Surprisingly, we discover that one can calculate the expectation or ensemble of this ELBO without any additional computational overhead. This finding allows us to design a new diffusion classifier that functions as an ensemble of EPNDC but does not require extra computational cost. We refer to this enhanced diffusion classifier as the Approximated Posterior Noised Diffusion Classifier (APNDC). Towards the end of this paper, we reduce the time complexity of diffusion classifiers by significantly decreasing variance through the use of the same noisy samples for all classes and by proposing a search algorithm to narrow down the candidate classes for the diffusion classifier.

Experimental results substantiate the superior performance of our methods. Notably, we achieve 82.2%, 70.7%, and 54.5% at $\ell_2$ radii of 0.25, 0.5, and 0.75, respectively, on the CIFAR-10 dataset. These results surpass the previous state-of-the-art [47] by absolute margins of 5.6%, 6.1%, and 4.1% in the corresponding categories. Additionally, our approach registers a clean accuracy of 91.2%, outperforming Xiao et al. [47] by 3.6%. Moreover, our time complexity reduction techniques decrease the computational burden by a factor of 10 on CIFAR-10 and by a factor of 1000 on ImageNet, without compromising certified robustness. Furthermore, our comparative analysis with heuristic methods not only highlights the tangible benefits of our theoretical advancements but also provides valuable insights into several evidence lower bounds and the inherent robustness of diffusion classifiers.

The contributions of this paper are summarized as follows:

- We derive the Lipschitz constant and certified lower bound for diffusion classifiers, demonstrating their inherent provable robustness.
- We generalize diffusion classifiers to classify noisy data, enabling us to derive non-constant Lipschitzness and state-of-the-art certified robustness.
- We propose a variance reduction technique that greatly reduces time complexity without compromising certified robustness.

## 2 Background

### 2.1 Diffusion Models

For simplicity in derivation, we introduce a general formulation that covers various diffusion models. In Appendix A.8, we show that common models, such as Ho et al. [11], Song et al. [42], Kingma et al. [17] and Karras et al. [16], can be transformed to align with our definition.

Given $\mathbf{x} := \mathbf{x}_0 \in [0, 1]^D$ with a data distribution $q(\mathbf{x}_0)$, the forward diffusion process incrementally introduces Gaussian noise to the data distribution, resulting in a continuous sequence of distributions $\{q(\mathbf{x}_t) := q_t(\mathbf{x}_t)\}_{t=1}^T$ by:

$$q(\mathbf{x}_t) = \int q(\mathbf{x}_0)q(\mathbf{x}_t|\mathbf{x}_0)d\mathbf{x}_0, \tag{1}$$

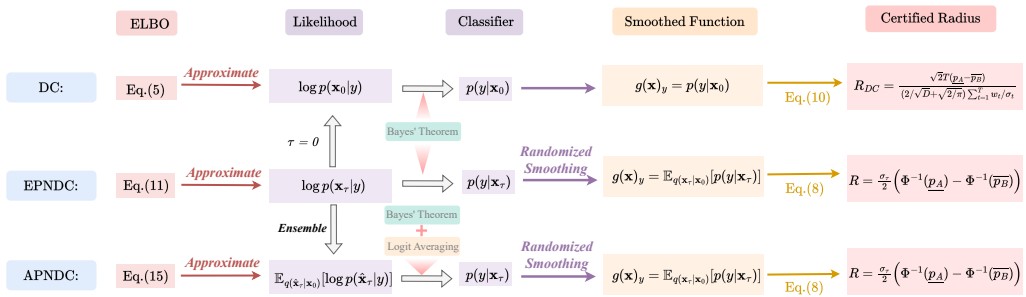

Figure 1: Illustration of our theoretical contributions. We derive the Lipschitz constant and the corresponding certified radius for diffusion classifiers [3]. Additionally, we introduce two novel evidence lower bounds, which are used to approximate the log likelihood. These lower bounds are then employed to construct classifiers based on Bayes' theorem. By applying randomized smoothing to these classifiers, we derive their certified robust radii.

where $q(\mathbf{x}_t|\mathbf{x}_0) = \mathcal{N}(\mathbf{x}_t; \mathbf{x}_0, \sigma_t^2 \mathbf{I})$, i.e., $\mathbf{x}_t = \mathbf{x}_0 + \sigma_t \boldsymbol{\epsilon}, \boldsymbol{\epsilon} \sim \mathcal{N}(\mathbf{0}, \mathbf{I})$. Typically, $\sigma_t$ monotonically increases with $t$, establishing one-to-one mappings $t(\sigma)$ from $\sigma$ to $t$ and $\sigma(t)$ from $t$ to $\sigma$. Additionally, $\sigma_T$ is large enough that $q(\mathbf{x}_T)$ is approximately an isotropic Gaussian distribution. Given $p := p_\theta$ as the parameterized reverse distribution with prior $p(\mathbf{x}_T) = \mathcal{N}(\mathbf{x}_T; \mathbf{0}, \sigma_T^2 \boldsymbol{I})$, the diffusion process used to synthesize real data is defined as a Markov chain with learned Gaussian distributions [11, 42]:

$$p(\mathbf{x}_{0:T}) = p(\mathbf{x}_T) \prod_{t=1}^{T} p(\mathbf{x}_{t-1}|\mathbf{x}_t). \tag{2}$$

In this work, we parameterize the reverse Gaussian distribution $p(\mathbf{x}_{t-1}|\mathbf{x}_t)$ using a neural network $\mathbf{h}_\theta(\mathbf{x}_t, t)$ as

$$p(\mathbf{x}_{t-1}|\mathbf{x}_t) = \mathcal{N}(\mathbf{x}_{t-1}; \boldsymbol{\mu}_\theta(\mathbf{x}_t, t), \frac{\sigma_t^2(\sigma_{t+1}^2 - \sigma_t^2)}{\sigma_{t+1}^2}\mathbf{I}),$$
$$\boldsymbol{\mu}_\theta(\mathbf{x}_t, t) = \frac{(\sigma_t^2 - \sigma_{t-1}^2)\mathbf{h}_\theta(\mathbf{x}_t, \sigma_t) + \sigma_{t-1}^2 \mathbf{x}_t}{\sigma_t^2}. \tag{3}$$

The parameter $\theta$ is usually trained by optimizing the evidence lower bound (ELBO) on the log likelihood [41, 11, 17]:

$$\log p(\mathbf{x}_0) \geq -\sum_{t=1}^{T} \mathbb{E}_{\boldsymbol{\epsilon}}\left[w_t \|\mathbf{h}_\theta(\mathbf{x}_t, \sigma_t) - \mathbf{x}_0\|_2^2\right] + C_1, \tag{4}$$

where $w_t = \frac{\sigma_{t+1} - \sigma_t}{\sigma_{t+1}^3}$ is the weight of the loss at time step $t$ and $C_1$ is a constant. Similarly, the conditional diffusion model $p(\mathbf{x}_{t-1}|\mathbf{x}_t, y)$ is parameterized by $\mathbf{h}_\theta(\mathbf{x}_t, \sigma_t, y)$. A similar lower bound on conditional log likelihood is

$$\log p(\mathbf{x}_0|y) \geq -\sum_{t=1}^{T} \mathbb{E}_{\boldsymbol{\epsilon}}\left[w_t \|\mathbf{h}_\theta(\mathbf{x}_t, \sigma_t, y) - \mathbf{x}_0\|_2^2\right] + C, \tag{5}$$

where $C$ is another constant.

## 2.2 Diffusion Classifiers

Diffusion classifier [3, 5, 26] $\text{DC}(\cdot) : [0, 1]^D \to \mathbb{R}^K$ is a generative classifier that uses a single off-the-shelf diffusion model for robust classification. It first approximates the conditional likelihood $\log p(y|\mathbf{x}_0)$ via conditional ELBO (i.e., using ELBO as logit), and then calculates the class probability $p(y|\mathbf{x}_0) \propto p(\mathbf{x}_0|y)$ through Bayes' theorem, with the assumption that $p(y)$ is a uniform prior:

$$\text{DC}(\mathbf{x}_0)_y := \frac{\exp(-\frac{1}{DT}\sum_{t=1}^{T} \mathbb{E}_{\boldsymbol{\epsilon}}\left[w_t \|\mathbf{h}_\theta(\mathbf{x}_t, \sigma_t, y) - \mathbf{x}_0\|_2^2\right])}{\sum_{\hat{y}} \exp\left(-\frac{1}{DT}\sum_{t=1}^{T} \mathbb{E}_{\boldsymbol{\epsilon}}\left[w_t \|\mathbf{h}_\theta(\mathbf{x}_t, \sigma_t, \hat{y}) - \mathbf{x}_0\|_2^2\right]\right)}$$
$$\approx \frac{\exp(\log p(\mathbf{x}_0|y))}{\sum_{\hat{y}} \exp\left(\log p(\mathbf{x}_0|\hat{y})\right)} = \frac{p(\mathbf{x}_0|y)p(y)}{\sum_{\hat{y}} p(\mathbf{x}_0|\hat{y})p(\hat{y})} \triangleq p(y|\mathbf{x}_0). \tag{6}$$

In other words, it utilizes the ELBO of each conditional likelihood $\log p(y|\mathbf{x}_0)$ as the logit of each class. This classifier achieves state-of-the-art empirical robustness across several types of threat models and can generalize to unseen attacks as it does not require training on adversarial examples [3]. However, there is still lacking a rigorous theoretical analysis, leaving questions about whether they will be vulnerable to (potentially) future stronger adaptive attacks.

## 2.3 Randomized Smoothing

Randomized smoothing [23, 6, 48, 20] is a model-agnostic technique designed to establish a lower bound of robustness against adversarial examples. It is scalable to large networks and datasets and achieves state-of-the-art performance in certified robustness [48]. This approach constructs a smoothed classifier by averaging the output of a base classifier over Gaussian noise. Owing to the Lipschitz continuity of this classifier, it remains stable within a certain perturbation range, thereby ensuring certified robustness.

Formally, given a classifier $f : [0, 1]^D \rightarrow \mathbb{R}^K$ that takes a $D$-dimensional input $\mathbf{x}_0$ and predicts class probabilities over $K$ classes, the $y$-th output of the smoothed classifier $g$ is:

$$g(\mathbf{x}_0)_y = P(\underset{\hat{y}\in\{1,...,K\}}{\arg\max} f(\mathbf{x}_0 + \sigma_\tau \cdot \boldsymbol{\epsilon})_{\hat{y}} = y), \tag{7}$$

where $\boldsymbol{\epsilon} \sim \mathcal{N}(\mathbf{0}, \mathbf{I})$ is a Gaussian noise and $\sigma_\tau$ is the noise level. Let $\Phi^{-1}$ denote the inverse function of the standard Gaussian CDF. Salman et al. [38] prove that $\Phi^{-1}(g(\mathbf{x}_0)_y)$ is $\frac{1}{\sigma_\tau}$-Lipschitz. However, the exact computation of $g(\mathbf{x}_0)$ is infeasible due to the challenge of calculating the expectation in a high-dimensional space. Practically, one usually estimates a lower bound $\underline{p_A}$ of $g(\mathbf{x}_0)_y$ and an upper bound $\overline{p_B}$ of $\max_{\hat{y}\neq y} g(\mathbf{x}_0)_{\hat{y}}$ using the Clopper-Pearson lemma, and then calculates the lower bound of the certified robust radius $R$ for class $y$ as

$$R = \frac{\sigma_\tau}{2}\left(\Phi^{-1}(\underline{p_A}) - \Phi^{-1}(\overline{p_B})\right). \tag{8}$$

Typically, the existing classifiers are trained to classify images in clean distribution $q(\mathbf{x}_0)$. However, the input distribution in Eq. (7) is $q(\mathbf{x}_\tau) = \int q(\mathbf{x}_0)q(\mathbf{x}_\tau|\mathbf{x}_0)d\mathbf{x}_0$. Due to the distribution discrepancy, $g(\mathbf{x}_0)$ constructed by classifiers trained on clean distribution $q(\mathbf{x}_0)$ exhibits low accuracy on $q(\mathbf{x}_\tau)$. Due to this issue, we cannot directly incorporate the diffusion classifier [3] with randomized smoothing. In this paper, we propose a new category of diffusion classifiers, that can directly calculate $p(y|\mathbf{x}_\tau)$ via an off-the-shelf diffusion model.

To handle Gaussian-corrupted data, early work [6, 38] trains new classifiers on $q(\mathbf{x}_\tau)$ but is not applicable to pre-trained models. Orthogonal to our work, there is also some recent work on using denoiser for certified robustness [39, 46], and some of them choose diffusion model as denoiser [2, 47, 52]. They first denoise $\mathbf{x}_\tau \sim q(\mathbf{x}_\tau)$, followed by an off-the-shelf discriminative classifier for classifying the denoised image. However, the efficacy of such an algorithm is largely constrained by the performance of the discriminative classifier.

## 3 Methodology

In this section, we first derive an upper bound of the Lipschitz constant in Sec. 3.1. Due to the difficulty in deriving a tighter Lipschitzness for such a mathematical form, we propose two variants of Noised Diffusion Classifiers (NDCs), and integrate them with randomized smoothing to obtain a tighter robust radius, as detailed in Sec. 3.2 and Sec. 3.3. Finally, we propose several techniques in Sec. 3.4 to reduce time complexity and enhance scalability for large datasets.

## 3.1 The Lipschitzness of Diffusion Classifiers

We observe that the logits $-\frac{1}{DT}\sum_{t=1}^{T} w_t \mathbb{E}_{\boldsymbol{\epsilon}}\left[\|\mathbf{h}_\theta(\mathbf{x}_t, \sigma_t) - \mathbf{x}_0\|_2^2\right]$ of diffusion classifier in Eq. (6) can be decomposed as

$$-\frac{1}{DT}\sum_{t=1}^{T} w_t \left(\mathbb{E}_{\boldsymbol{\epsilon}}\left[\|\mathbf{h}_\theta(\mathbf{x}_t, \sigma_t)\|_2^2\right] + \|\mathbf{x}_0\|_2^2 - 2\mathbb{E}_{\boldsymbol{\epsilon}}\left[\mathbf{h}_\theta(\mathbf{x}_t, \sigma_t)^\top\right]\mathbf{x}_0\right). \tag{9}$$

---

**Algorithm 1** EPNDC

---

1: **Require:** A pre-trained diffusion model $\mathbf{h}_\theta$, a noisy input image $\mathbf{x}_\tau$, noisy level $\tau$.
2: **for** $y = 0$ to $K - 1$ **do**
3:     **for** $t = \tau + 1$ **to** $T$ **do**
4:         Calculate the analytical form of $\mathbb{E}_{q(\mathbf{x}_t|\mathbf{x}_{t+1},\mathbf{x}_\tau,y)}[\mathbf{x}_t]$ by: $\frac{(\sigma_{t+1}^2-\sigma_t^2)\mathbf{x}_\tau+(\sigma_t^2-\sigma_\tau^2)\mathbf{x}_{t+1}}{\sigma_{t+1}^2-\sigma_\tau^2}$
5:         Calculate the analytical form of $\mathbb{E}_{p(\mathbf{x}_t|\mathbf{x}_{t+1},y)}[\mathbf{x}_t]$ by: $\frac{(\sigma_{t+1}^2-\sigma_t^2)\mathbf{h}(\mathbf{x}_{t+1},\sigma_{t+1},y)+\sigma_t^2\mathbf{x}_{t+1}}{\sigma_{t+1}^2}$
6:     **end for**
7:     Calculate the lower bound $\underline{\log p(\mathbf{x}_\tau|y)}$ of $\log p(\mathbf{x}_\tau|y)$ using $w_t^{(\tau)} = \frac{\sigma_{t+1}^2-\sigma_\tau^2}{2(\sigma_t^2-\sigma_\tau^2)(\sigma_{t+1}^2-\sigma_t^2)}$ by:
    $\sum_{t=\tau+1}^T w_t^{(\tau)}\mathbb{E}_{q(\mathbf{x}_{t+1}|\mathbf{x}_\tau)}\|\mathbb{E}_{q(\mathbf{x}_t|\mathbf{x}_{t+1},\mathbf{x}_\tau,y)}[\mathbf{x}_t] - \mathbb{E}_{p(\mathbf{x}_t|\mathbf{x}_{t+1},y)}[\mathbf{x}_t]\|^2$;
8: **end for**
9: Approximate $p_\theta(y|\mathbf{x}_\tau)$ by $\frac{\exp(\underline{\log p_\theta(\mathbf{x}_\tau|y)})}{\sum_{\hat{y}}\exp(\underline{\log p_\theta(\mathbf{x}_\tau|\hat{y})})}$, and **Return:** $\tilde{y} = \arg\max_y p_\theta(y|\mathbf{x}_\tau)$.

---

Given that $\mathbb{E}_\epsilon\left[\|\mathbf{h}_\theta(\mathbf{x}_t,\sigma_t)\|_2^2\right]$ and $\mathbb{E}_\epsilon\left[\mathbf{h}_\theta(\mathbf{x}_t,\sigma_t)\right]$ are smoothed by Gaussian noise and lie within the range $[0,1]^D$, they satisfy the Lipschitz condition [38]. Consequently, the logits of diffusion classifiers should satisfy a Lipschitz condition. Thus, it can be inferred that the entire diffusion classifier is robust and possesses a certain robust radius.

We derive an upper bound for the Lipchitz constant of diffusion classifiers in the following theorem:

**Theorem 3.1.** *The upper bound of Lipschitz constant of diffusion classifier is given by:*

$$|DC(\mathbf{x}_0+\boldsymbol{\delta})_y - DC(\mathbf{x}_0)_y| \le \frac{1}{2\sqrt{2}}\sum_{t=1}^T \frac{w_t}{\sigma_t T}\left(\sqrt{\frac{2}{\pi}}+\frac{2}{\sqrt{D}}\right)\|\boldsymbol{\delta}\|_2. \tag{10}$$

*If one can get a lower bound $\underline{p_A}$ for $DC(\mathbf{x}_0)_y$ and a upper bound $\overline{p_B}$ for $\max_{\hat{y}\neq y} DC(\mathbf{x}_0)_{\hat{y}}$ (e.g., probabilistic bound by Bernstein inequality [30]), the lower bound of certified radius for diffusion classifier can be obtained:*

$$R_{DC} = \frac{\sqrt{2}T(\underline{p_A}-\overline{p_B})}{(2/\sqrt{D}+\sqrt{2/\pi})\sum_{t=1}^T w_t/\sigma_t}. \tag{11}$$

*Proof.* (Sketch; details in Appendix A.2). Employing a similar methodology to that used by Salman et al. [38], we derive the gradient of the diffusion classifier. Since the gradient norm of a neural network is unbounded, we transfer the target of the gradient operator from the neural network to the Gaussian density function, so that we can bound the gradient norm and the Lipschitz constant. ☐

As demonstrated in Theorem 3.1, the Lipschitz constant of diffusion classifiers is nearly identical to that in the "*weak law*" of randomized smoothing (See Appendix A.3, or Lemma 1 in Salman et al. [38]). This constant is small and independent of the dimension $D$, indicating the inherent robustness of diffusion classifiers. However, similar to the weak law of randomized smoothing, such certified robustness has limitations because it assumes the maximum Lipschitz condition is satisfied throughout the entire perturbation path, i.e., it assumes the equality always holds in $|f(\mathbf{x}_{adv})_y - f(\mathbf{x})_y| \le L\|\mathbf{x}-\mathbf{x}_{adv}\|_2$ when $f$ has Lipschitz constant $L$. As a result, the equality also holds in $f(\mathbf{x}_{adv})_y \ge f(\mathbf{x})_y - L\|\mathbf{x}-\mathbf{x}_{adv}\|_2$ and $f(\mathbf{x}_{adv})_{\hat{y}} \le f(\mathbf{x})_{\hat{y}} + L\|\mathbf{x}-\mathbf{x}_{adv}\|_2$ for $\max_{\hat{y}\neq y} f(\mathbf{x})_{\hat{y}}$. To guarantee the prediction is unchanged (i.e., $f(\mathbf{x}_{adv})_y \ge f(\mathbf{x}_{adv})_{\hat{y}}$), its requires the perturbation $\|\mathbf{x}-\mathbf{x}_{adv}\|_2$ must be less than $\frac{1}{2L}$.

To be specific, under the weak law of randomized smoothing, it is impossible to achieve a certified radius greater than 1.253. According to Eq. (11), a certified radius greater than 0.39 is unattainable, and empirically, the average certified radius achieved is only 0.156 (refer to Appendix C.3). This is significantly lower than the empirical robustness upper bound obtained through adaptive attacks as reported in Chen et al. [3].

On the other hand, the "*strong law*" of randomized smoothing (See Eq. (8) or Lemma 2 in Salman et al. [38]) can yield a non-constant Lipschitzness, leading to a more precise robust radius, with the upper bound of the certified radius potentially being infinite. Therefore, in the subsequent sections, we aim to combine diffusion classifier with randomized smoothing to achieve a tighter certified radius, thus thoroughly explore its robustness.

---
**Algorithm 2** APNDC
---
1: **Require:** A pre-trained diffusion model $\mathbf{h}_\theta$, a noisy input image $\mathbf{x}_\tau$, noisy level $\tau$.
2: **for** $y = 0$ **to** $K - 1$ **do**
3:     Calculate the lower bound $\underline{\log p(\mathbf{x}_\tau | y)}$ of $\log p(\mathbf{x}_\tau | y)$ by:
$$\sum_{t=\tau+1}^{T} w_t \|\mathbf{h}_\theta(\mathbf{x}_\tau, \sigma_\tau) - \mathbf{h}_\theta(\mathbf{x}_t, \sigma_{t+1}, y)\|^2, \text{ where } w_t = \frac{\sigma_{t+1} - \sigma_t}{\sigma_{t+1}^3};$$
4: **end for**
5: Approximate $p_\theta(y|\mathbf{x}_\tau)$ by $\frac{\exp\left(\underline{\log p_\theta(\mathbf{x}_\tau|y)}\right)}{\sum_{\hat{y}} \exp\left(\underline{\log p_\theta(\mathbf{x}_\tau|\hat{y})}\right)}$;
6: **Return:** $\tilde{y} = \arg\max_y p_\theta(y|\mathbf{x}_\tau)$.
---

## 3.2 Exact Posterior Noised Diffusion Classifier

As explained in Sec. 2.3, randomized smoothing constructs a smoothed classifier $g$ from a given base classifier $f$ by aggregating votes over Gaussian-corrupted data. This process necessitates that the base classifier can classify data from the noisy distribution $q(\mathbf{x}_\tau)$. However, the diffusion classifier in Chen et al. [3] is limited to classifying data solely from $q(\mathbf{x}_0)$. Therefore, in this section, we generalize the diffusion classifier to enable the classification of images from $q(\mathbf{x}_\tau)$ for any given $\tau$.

Similar to Chen et al. [3], our fundamental idea involves deriving the ELBO for $\log p(\mathbf{x}_\tau|y)$ and subsequently calculating $p(y|\mathbf{x}_\tau)$ using the estimated $\log p(\mathbf{x}_\tau|y)$ via Bayes' theorem. Drawing inspiration from Ho et al. [11], we derive a similar ELBO for $\log p(\mathbf{x}_\tau)$, as elaborated in the following theorem (the conditional ELBO is similar to unconditional one, see Appendix A.4 for details):

**Theorem 3.2.** *(Proof in Appendices A.4 and A.5). The ELBO of $\log p(\mathbf{x}_\tau)$ is given by:*

$$\log p(\mathbf{x}_\tau) \geq -\sum_{t=\tau}^{T} w_t^{(\tau)} \mathbb{E}_{q(\mathbf{x}_{t+1}|\mathbf{x}_\tau)}\left[\|\mathbb{E}_{q(\mathbf{x}_t|\mathbf{x}_{t+1},\mathbf{x}_\tau)}[\mathbf{x}_t] - \mathbb{E}_{p(\mathbf{x}_t|\mathbf{x}_{t+1})}[\mathbf{x}_t]\|^2\right] + C_2, \quad (12)$$

*where*

$$\mathbf{x}_{t+1} \sim q(\mathbf{x}_{t+1}|\mathbf{x}_\tau), \quad \mathbb{E}_{q(\mathbf{x}_t|\mathbf{x}_{t+1},\mathbf{x}_\tau)}[\mathbf{x}_t] = \frac{(\sigma_{t+1}^2 - \sigma_t^2)\mathbf{x}_\tau + (\sigma_t^2 - \sigma_\tau^2)\mathbf{x}_{t+1}}{\sigma_{t+1}^2 - \sigma_\tau^2},$$

$$w_t^{(\tau)} = \frac{\sigma_{t+1}^2 - \sigma_\tau^2}{2(\sigma_t^2 - \sigma_\tau^2)(\sigma_{t+1}^2 - \sigma_t^2)}, \quad \mathbb{E}_{p(\mathbf{x}_t|\mathbf{x}_{t+1})}[\mathbf{x}_t] = \frac{(\sigma_{t+1}^2 - \sigma_t^2)h(\mathbf{x}_{t+1}, \sigma_{t+1}) + \sigma_t^2 \mathbf{x}_{t+1}}{\sigma_{t+1}^2}. \quad (13)$$

*Remark* 3.3. Notice that the summation of KL divergence in the ELBO of $\log p(\mathbf{x}_\tau)$ starts from $\tau + 1$ and ends at $T$, while that of $\log p(\mathbf{x}_0)$ starts from 1. Besides, the posterior is $q(\mathbf{x}_t|\mathbf{x}_{t+1}, \mathbf{x}_\tau)$ instead of $q(\mathbf{x}_t|\mathbf{x}_{t+1}, \mathbf{x}_0)$.

*Remark* 3.4. When $\tau = 0$, this result degrades to the diffusion training loss $\frac{\sigma_{t+1} - \sigma_t}{\sigma_{t+1}^3}\|\mathbf{x}_0 - \mathbf{h}(\mathbf{x}_{t+1}, \sigma_{t+1})\|^2$, consistent with Kingma et al. [17] and Karras et al. [16].

Due to the page width limit, we only present the unconditional ELBO in the main text. We can get the conditional ELBO by adding $y$ to the condition. Using the conditional ELBOs as approximation for log likelihood (i.e., using ELBOs as logits), one can calculate $p(y|\mathbf{x}_\tau) = \frac{e^{\log p_\theta(\mathbf{x}_\tau|y)}}{\sum_{\hat{y}} e^{\log p_\theta(\mathbf{x}_\tau|\hat{y})}}$ for classification. We name this algorithm as Exact Posterior Noised Diffusion Classifier (EPNDC), as demonstrated in Algorithm 1.

Although this classifier achieves non-trivial certified robustness, it still has limitations. For instance, we cannot theoretically determine the optimal weight $w_t^{(\tau)}$ (see Appendix C.2 for details). Additionally, the time complexity is high. In the next section, we propose a new diffusion classifier called the Approximated Posterior Noised Diffusion Classifier (APNDC), which addresses these problems and acts like an ensemble of EPNDC so that greatly enhanced certified robustness without any computational overhead.

## 3.3 Approximated Posterior Noised Diffusion Classifier

Greatly inspired by Song et al. [42] and Meng et al. [31], we propose to approximate the posterior in a similar manner:

$$q(\mathbf{x}_t|\mathbf{x}_{t+1}, \mathbf{x}_\tau) = q(\mathbf{x}_t|\mathbf{x}_{t+1}, \mathbf{x}_\tau, \mathbf{x}_0 = \mathbf{h}_\theta(\mathbf{x}_\tau, \sigma_\tau)) \approx q(\mathbf{x}_t|\mathbf{x}_{t+1}, \mathbf{x}_0 = \mathbf{h}_\theta(\mathbf{x}_\tau, \sigma_\tau)). \quad (14)$$

Table 1: Certified accuracy at CIFAR-10 test set. The clean accuracy for each smoothed model is in the parentheses. The certified accuracy for each cell is from Xiao et al. [47], same as the results from their respective papers. Carlini et al. [2] and Xiao et al. [47] use ImageNet-21k as extra data.

| Method | Off-the-shelf | Extra data | Certified Accuracy at $R$ (%) | | | |
|---|---|---|---|---|---|---|
| | | | 0.25 | 0.5 | 0.75 | 1.0 |
| PixelDP [23] | ✗ | ✗ | $^{(71.0)}22.0$ | $^{(44.0)}2.0$ | - | - |
| RS [6] | ✗ | ✗ | $^{(75.0)}61.0$ | $^{(75.0)}43.0$ | $^{(65.0)}32.0$ | $^{(65.0)}23.0$ |
| SmoothAdv [38] | ✗ | ✗ | $^{(82.0)}68.0$ | $^{(76.0)}54.0$ | $^{(68.0)}41.0$ | $^{(64.0)}32.0$ |
| Consistency [13] | ✗ | ✗ | $^{(77.8)}68.8$ | $^{(75.8)}58.1$ | $^{(72.9)}48.5$ | $^{(52.3)}37.8$ |
| MACER [50] | ✗ | ✗ | $^{(81.0)}71.0$ | $^{(81.0)}59.0$ | $^{(66.0)}46.0$ | $^{(66.0)}38.0$ |
| Boosting [12] | ✗ | ✗ | $^{(83.4)}70.6$ | $^{(76.8)}60.4$ | $^{(71.6)}52.4$ | $^{(73.0)}\mathbf{38.8}$ |
| SmoothMix [14] | ✓ | ✗ | $^{(77.1)}67.9$ | $^{(77.1)}57.9$ | $^{(74.2)}47.7$ | $^{(61.8)}37.2$ |
| Denoised [39] | ✓ | ✗ | $^{(72.0)}56.0$ | $^{(62.0)}41.0$ | $^{(62.0)}28.0$ | $^{(44.0)}19.0$ |
| Lee [24] | ✓ | ✗ | $^{(-)}60.0$ | $^{(-)}42.0$ | $^{(-)}28.0$ | $^{(-)}19.0$ |
| Carlini [2] | ✓ | ✓ | $^{(88.0)}73.8$ | $^{(88.0)}56.2$ | $^{(88.0)}41.6$ | $^{(74.2)}31.0$ |
| DensePure [47] | ✓ | ✓ | $^{(87.6)}76.6$ | $^{(87.6)}64.6$ | $^{(87.6)}50.4$ | $^{(73.6)}37.4$ |
| DiffPure+DC (baseline, ours) | ✓ | ✗ | $^{(87.5)}68.8$ | $^{(87.5)}53.1$ | $^{(87.5)}41.2$ | $^{(73.4)}25.6$ |
| EPNDC ($T' = 100$, ours) | ✓ | ✗ | $^{(89.1)}77.4$ | $^{(89.1)}60.0$ | $^{(89.1)}35.7$ | $^{(74.8)}24.4$ |
| APNDC ($T' = 100$, ours) | ✓ | ✗ | $^{(89.5)}80.7$ | $^{(89.5)}68.8$ | $^{(89.5)}50.8$ | $^{(76.2)}35.2$ |
| APNDC ($T' = 1000$, ours) | ✓ | ✗ | $^{(\mathbf{91.2})}\mathbf{82.2}$ | $^{(\mathbf{91.2})}\mathbf{70.7}$ | $^{(\mathbf{91.2})}\mathbf{54.5}$ | $^{(\mathbf{77.3})}38.2$ |

As a result, the KL divergence can be simplified using this approximation:

$$D_{\mathrm{KL}}(q(\mathbf{x}_t|\mathbf{x}_{t+1},\mathbf{x}_\tau)\|p_\theta(\mathbf{x}_t|\mathbf{x}_{t+1})) \approx D_{\mathrm{KL}}(q(\mathbf{x}_t|\mathbf{x}_{t+1},\mathbf{x}_0 = \mathbf{h}_\theta(\mathbf{x}_\tau,\sigma_\tau))\|p_\theta(\mathbf{x}_t|\mathbf{x}_{t+1}))$$
$$=\frac{\sigma_{t+1}-\sigma_t}{\sigma_{t+1}^3}\|\mathbf{h}(\mathbf{x}_\tau,\sigma_\tau) - \mathbf{h}(\mathbf{x}_{t+1},\sigma_{t+1})\|^2 + C_3. \tag{15}$$

Intriguingly, Eq. (15) is the ELBO of $\mathbb{E}_{q(\hat{\mathbf{x}}_\tau|\mathbf{x}_0=\mathbf{h}_\theta(\mathbf{x}_\tau,\sigma_\tau))}[\log p_\tau(\hat{\mathbf{x}}_\tau)]$, i.e.,

$$\mathbb{E}_{q(\hat{\mathbf{x}}_\tau|\mathbf{x}_0=\mathbf{h}_\theta(\mathbf{x}_\tau,\sigma_\tau))}[\log p_\tau(\hat{\mathbf{x}}_\tau)] \geq C_4 - \sum_{t=\tau+1}^{T-1} w_t \mathbb{E}_{q(\mathbf{x}_t|\mathbf{x}_0=\mathbf{h}_\theta(\mathbf{x}_\tau,\sigma_\tau))}[\|\mathbf{h}_\theta(\mathbf{x}_t,\sigma_t) - \mathbf{x}_0\|_2^2]. \tag{16}$$

Therefore, one can use the ELBO in Eq. (16) as a approximation for $\log p(\mathbf{x}_\tau)$ (i.e., employing the ELBOs of this expected log likelihood as the logits), and calculate the class probabilities through Bayes' theorem. We name this method as Approximated Posterior Noised Diffusion Classifier (APNDC), as shown in Algorithm 2.

APNDC is functionally equivalent to an ensemble of EPNDC, as it calculates the ELBO of $\mathbb{E}_{q(\hat{\mathbf{x}}_\tau|\mathbf{x}_0=\mathbf{h}_\theta(\mathbf{x}_\tau,\sigma_\tau))}[\log p_\tau(\hat{\mathbf{x}}_\tau)]$, which corresponds to the expected $\log p(\mathbf{x}_\tau|y)$. This nearly-free ensemble can be executed with only one more forward pass of UNet to compute $\mathbf{h}_\theta(\mathbf{x}_\tau,\tau)$. For detailed explanations, please refer to Appendix A.7.

*Remark* 3.5. From a heuristic standpoint, one might consider first employing a diffusion model for denoising (named DiffPure by Nie et al. [34]), followed by using a diffusion classifier for classification. This approach differs from our method, where we calculate the diffusion loss only from $\tau + 1$ to $T$, and the noisy samples $\mathbf{x}_t$ are obtained by adding noise to $\mathbf{x}_\tau$ instead of $\mathbf{x}_0$. In Table 1, we demonstrate that our APNDC method significantly outperforms this heuristic approach (DiffPure+DC).

### 3.4 Time Complexity Reduction

**Variance reduction.** The main computational effort in our approach is dedicated to calculating the evidence lower bound for each class. This involves computing the sum of reconstruction losses. For instance, in DC, the reconstruction loss is $\|\mathbf{x}_0 - \mathbf{h}_\theta(\mathbf{x}_t,\sigma_{t+1})\|_2^2$. In EPNDC, it is $\|\mathbb{E}_{q(\mathbf{x}_t|\mathbf{x}_{t+1},\mathbf{x}_\tau)}[\mathbf{x}_t] - \mathbb{E}_{p(\mathbf{x}_t|\mathbf{x}_{t+1})}[\mathbf{x}_t]\|_2^2$, and in APNDC, the loss is $\|\mathbf{h}_\theta(\mathbf{x}_\tau,\sigma_\tau) - \mathbf{h}_\theta(\mathbf{x}_{t+1},\sigma_{t+1})\|_2^2$, with the summation carried out over $t$. Chen et al. [3] attempt to reduce the time complexity by only calculating the reconstruction loss at certain timesteps. However, this approach proves ineffective. We identify that the primary reason for this failure is the large variance in the reconstruction

Table 2: Certified accuracy at ImageNet-64x64. The clean accuracy is in the parentheses.

| Method | Off-the-shelf | Extra data | Certified Accuracy at $R$ (%) | | | |
|---|---|---|---|---|---|---|
| | | | 0.25 | 0.5 | 0.75 | 1.0 |
| RS [6] | ✗ | ✗ | $^{(45.5)}$37.3 | $^{(45.5)}$26.6 | $^{(37.0)}$20.9 | $^{(37.0)}$15.1 |
| SmoothAdv [38] | ✗ | ✗ | $^{(44.4)}$37.4 | $^{(44.4)}$27.9 | $^{(34.7)}$21.1 | $^{(34.7)}$17.0 |
| Consistency [13] | ✗ | ✗ | $^{(43.6)}$36.9 | $^{(43.6)}$31.5 | $^{(43.6)}$26.0 | $^{(31.4)}$16.6 |
| MACER [50] | ✗ | ✗ | $^{(46.3)}$35.7 | $^{(46.3)}$27.1 | $^{(46.3)}$15.6 | $^{(38.7)}$11.3 |
| Carlini [2] | ✓ | ✗ | $^{(41.1)}$37.5 | $^{(39.4)}$30.7 | $^{(39.4)}$24.6 | $^{(39.4)}$21.7 |
| DensePure [47] | ✓ | ✗ | $^{(37.7)}$35.4 | $^{(37.7)}$29.3 | $^{(37.7)}$26.0 | $^{(37.7)}$18.6 |
| APNDC (Sift-and-Refine, ours) | ✓ | ✗ | $^{(54.4)}$**46.3** | $^{(54.4)}$**38.3** | $^{(43.5)}$**35.2** | $^{(43.5)}$**32.8** |

loss, necessitating sufficient calculations for convergence. To address this, we propose an effective variance reduction technique that uses identical input samples across all categories at each timestep. In other words, we use the same $\mathbf{x}_t$ for different classes. This approach significantly reduces the difference in prediction difficulty among various classes, allowing for a more equitable calculation of the reconstruction loss for each class. As shown in Figure 2(a), we can utilize a much smaller number of timesteps, such as 8, without sacrificing accuracy, thereby substantially reducing time complexity.

**Sift-and-refine algorithm.** The time complexity of these diffusion classifiers is proportional to the number of classes, presenting a significant obstacle for their application in datasets with numerous classes. Chen et al. [3] suggest the use of multi-head diffusion to address this issue. However, this approach requires training an additional diffusion model, leading to extra computational overhead. In our work, we focus solely on employing a single off-the-shelf diffusion model to construct a certifiably robust classifier. To tackle the aforementioned challenge, we propose a Sift-and-refine algorithm. The core idea is to swiftly reduce the number of classes, thereby limiting our focus to a manageable subset of classes. We provide more detailed analysis in Algorithm 5.

## 4 Experiment

Following previous studies [2, 47, 52], we evaluate the certified robustness of our method on two standard datasets, CIFAR-10 [19] and ImageNet [37], selecting a subset of 512 images from each. We adhere to the certified robustness pipeline established by Cohen et al. [6], although our method potentially offers a tighter certified bound, as demonstrated in Appendix A.3. To make a fair comparison with previous studies, we also select $\sigma_\tau \in \{0.25, 0.5, 1.0\}$ for certification (thus $\tau$ is determined) and use EDM [16] as our diffusion models. For a re-clarification on the hyper-parameters and additional experiments (including ablation studies on diffusion checkpoints and time complexity comparison), please refer to Appendix B.

### 4.1 Results on CIFAR-10

**Experimental settings.** Due to computational constraints, we employ a sample size of $N = 10,000$ to estimate $p_A$. The number of function evaluations (NFEs) for each image in our method is $O(N \cdot T' \cdot K)$, amounting to $10^8$ for $T' = 100$ and $10^9$ for $T' = 1000$ since $K = 10$ in this dataset. In contrast, the NFEs for the previous state-of-the-art method [47] are $4 \cdot 10^8$, which is four times higher than our method when $T'$ is 100. It is important to highlight that our sample size $N$ is 10 times smaller than those baselines in (Table 1), potentially placing our method at a significant disadvantage, especially for large $\sigma_\tau$.

**Experimental results.** As shown in Table 1, our method, utilizing an off-the-shelf model without the need for extra data, significantly outperforms all previous methods at smaller values of $\epsilon = \{0.25, 0.5\}$. Notably, it surpasses all previous methods on clean accuracy, and exceeds the previous state-of-the-art method [47] by 5.6% at $\epsilon = 0.25$ and 6.1% at $\epsilon = 0.5$. Even with larger values of $\epsilon$, our method attains performance levels comparable to existing approaches. This is particularly noteworthy considering our constrained setting of $N = 10,000$, substantially smaller than the $N = 100,000$ used in prior works. Considering that the community of randomized smoothing employs hypothesis tests to establish a probabilistic upper bound of the smoothed function, with consistent type-one error rates, our method encounters significant disadvantages. This is particularly the case since, with equivalent accuracy on noisy data, certified robustness is a monotonically

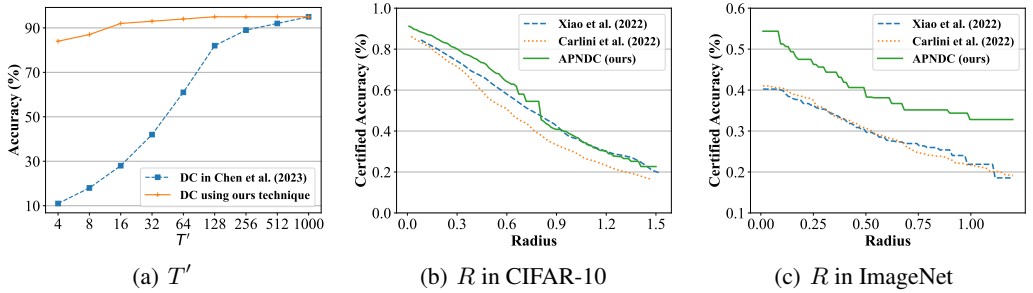

(a) $T'$      (b) $R$ in CIFAR-10      (c) $R$ in ImageNet

Figure 2: (a) The accuracy (%) on CIFAR-10 dataset with time complexity reduction technique in Chen et al. [3] and ours. (b, c) The upper envelop of certified radii of different methods.

increasing function with respect to sample size $N$. However, we still achieve competitive performance despite its inherent sample size disadvantage.

## 4.2 Results on ImageNet

**Experimental settings.** We conduct experiments on ImageNet64x64 due to the absence of conditional diffusion models for 256x256 resolution. Due to computational constraints, we employ a sample size of $N = 1000$, 10 times smaller than all other works in Table 2. We use the Sift-and-Refine algorithm to improve the efficiency.

**Experimental results.** As demonstrated in Table 2 and Figure 2(c), our method, only employing a single off-the-shelf diffusion model without requiring extra data, significantly outperforms previous training-based and training-free approaches. In contrast, diffusion-based purification methods, when applied with small CNNs and no extra data, do not maintain their superiority over training-based approaches. It is noteworthy that our experiments are conducted with only one-tenth of the sample size typically used in previous works. This success on a large dataset like ImageNet64x64 underscores the scalability of diffusion classifiers in handling extensive datasets with a larger number of classes.

## 4.3 Discussions

**Comparison with heuristic methods.** From a heuristic standpoint, one might consider initially using a diffusion model for denoising, followed by a diffusion classifier for classification. As shown in Table 1, this heuristic approach outperforms nearly all prior off-the-shelf and no-extra-data baselines. However, the methods derived through our theoretical analysis significantly surpass this heuristic strategy. This outcome underscores the practical impact of our theoretical contributions.

**Trivial performance of EPNDC.** Although EPNDC exhibits non-trivial improvements compared to previous methods, it still lags significantly behind APNDC. There are two main reasons for this gap. First, as extensively discussed in Appendix C.2, the weight in EPNDC is not optimal, and we cannot theoretically determine the optimal weight. Additionally, APNDC is equivalent to an ensemble of EPNDC, which may contribute to its superior performance compared to EPNDC.

**Explanation of Eq. (11).** Eq. (11) is extremely similar to *the weak law of randomized smoothing*. When $\sigma_t$ is larger, it could potentially have a larger certified radius, but the input images will be more noisy and hard to classify. This trade-off is quite similar to the role of $\sigma_\tau$ in randomized smoothing. However, there are two key differences. First, the inputs to the network contain different levels of noisy images, which means the network could see clean images, less noisy images, and very noisy images, hence could make more accurate predictions. Besides, the trade-off parameter is $\frac{w_t}{\sigma_t}$, allowing users some freedom to select different noise levels and balance them by $w_t$. We observe that this is the common feature of such denoising and reconstructing classifiers. We anticipate this observation will aid the community in developing more robust and certifiable defenses.

# 5 Conclusion

In this work, we conduct a comprehensive analysis of the robustness of diffusion classifiers. We establish their non-trivial Lipschitzness, a key factor underlying their remarkable empirical robustness. Furthermore, we extend the capabilities of diffusion classifiers to classify noisy data at any noise level by deriving the evidence lower bounds for noisy data distributions. This advancement enable us to combine the diffusion classifiers with randomized smoothing, leading to a tighter certified radius. Experimental results demonstrate substantial improvements in certified robustness and time complexity. We hope that our findings contribute to a deeper understanding of diffusion classifiers in the context of adversarial robustness and help alleviate concerns regarding their robustness.

## Acknowledgement

This work was supported by NSFC Projects (Nos. , 62076147, 92370124, 62350080, 92248303, 62276149, 62061136001), BNRist (BNR2022RC01006), Tsinghua Institute for Guo Qiang, and the High Performance Computing Center, Tsinghua University. J. Zhu was also supported by the XPlorer Prize. Y. Dong was also supported by the China National Postdoctoral Program for Innovative Talents and Shuimu Tsinghua Scholar Program.

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

**APPENDIX**
Appendix organization:

# A    Proofs

## A.1    Assumptions and Lemmas

**Assumption A.1.** We adopt the following assumptions. These assumptions are quite common in the context of certified robustness [38, 6] and diffusion models [42, 28], and they apply to most common neural networks:

1. Input image $\mathbf{x} \in [0,1]^D$.

2. $\forall 0 \le t \le T, q_t(\mathbf{x}) \in \mathcal{C}^2$, $\mathbf{h}(\mathbf{x}, \sigma_t) \in \mathcal{C}^1$ and $\mathbb{E}_{q_t(\mathbf{x})}[\|\mathbf{x}\|_2^2] \le \infty$.

3. For any classifier $f$ mentioned in this paper, $f(\mathbf{x}) \in \mathcal{C}^1$, and $\exists C \ge 0, \forall \mathbf{x} : \|f(\mathbf{x})\|_2 \le C$.

4. The output of the diffusion U-Net is bounded: $\mathbf{h}(\mathbf{x}, \sigma_t) \in [0,1]^D$ for all $\mathbf{x}$ and $t$. This property can be ensured by using the CLIP operator to clip the output of the U-Net.

**Lemma A.2.** *The second norm of the gradient of softmax function* $\|\frac{\partial}{\partial \mathbf{x}} softmax(\mathbf{x}/\beta)_y\|_2$ *is less than or equal to* $\frac{1}{2\sqrt{2}\beta}$. *In other words,*

$$\|\frac{\partial}{\partial \mathbf{x}} softmax(\mathbf{x}/\beta)_y\|_2 \le \frac{1}{2\sqrt{2}\beta}.$$

*Proof.*

$$
\begin{aligned}
& \max \|\frac{\partial}{\partial \mathbf{x}} \text{softmax}(\mathbf{x}/\beta)_y\|_2 \\
&= \max \|(\mathbf{e}_y - \text{softmax}(\mathbf{x}/\beta)) \frac{\text{softmax}(\mathbf{x}/\beta)_y}{\beta}\|_2 \\
&= \max \frac{1}{\beta} \cdot \sqrt{\text{softmax}(\mathbf{x}/\beta)_y^2 [(1 - \text{softmax}(\mathbf{x}/\beta)_y)^2 + \sum_{i \ne y} \text{softmax}(\mathbf{x}/\beta)_i^2]} \\
&\le \max \frac{\sqrt{2}}{\beta} \cdot \sqrt{\text{softmax}(\mathbf{x}/\beta)_y^2 (1 - \text{softmax}(\mathbf{x}/\beta)_y)^2} \le \frac{1}{2\sqrt{2}\beta}.
\end{aligned}
$$

$\square$

## A.2 Lipschitz Constant of Diffusion Classifiers

**Lemma A.3.** *(Adapted from Chen et al. [3].) The gradient of the diffusion classifier is given by*

$$\frac{d}{d\mathbf{x}}\mathbb{E}_{\boldsymbol{\epsilon}}[\|\mathbf{h}_\theta(\mathbf{x}_t, \sigma_t, y) - \mathbf{x}\|_2^2] = \mathbb{E}_{\boldsymbol{\epsilon}}[\frac{\partial \log p(\mathbf{x}_t|\mathbf{x})}{\partial \mathbf{x}}\|\mathbf{h}_\theta(\mathbf{x}_t, \sigma_t, y) - \mathbf{x}\|_2^2] + \mathbb{E}_{\boldsymbol{\epsilon}}[(\mathbf{h}_\theta(\mathbf{x}_t, \sigma_t, y) - \mathbf{x})\frac{2}{\sigma_t}],$$

*where*

$$\frac{\partial \log p(\mathbf{x}_t|\mathbf{x})}{\partial \mathbf{x}} = \frac{\partial}{\partial \mathbf{x}} \log \exp\left(-\frac{\|\mathbf{x}_t - \mathbf{x}\|_2^2}{2\sigma_t^2}\right) = -\frac{\mathbf{x} - \mathbf{x}_t}{\sigma_t^2} = \frac{\sigma_t\boldsymbol{\epsilon}}{\sigma_t^2} = \frac{\boldsymbol{\epsilon}}{\sigma_t}.$$

Lemma A.3 already derive the gradient of the diffusion classifier, and transfer the target of nabla operator to distribution function. Hence, we only need to bound the $\ell_2$ norm of both term in Lemma A.3.

For the first term:

$$\|\mathbb{E}_{\boldsymbol{\epsilon}}[\frac{\partial \log p(\mathbf{x}_t|\mathbf{x})}{\partial \mathbf{x}}\|\mathbf{h}_\theta(\mathbf{x}_t, \sigma_t, y) - \mathbf{x}\|_2^2]\|_2$$

$$=\|\int p(\boldsymbol{\epsilon})[\frac{\boldsymbol{\epsilon}}{\sigma_t}\|\mathbf{h}_\theta(\mathbf{x}_t, \sigma_t, y) - \mathbf{x}\|_2^2]d\boldsymbol{\epsilon}\|_2$$

$$\leq\frac{1}{\sigma_t}\max_{\|\mathbf{u}\|_2=1}\mathbf{u}^\top\int p(\boldsymbol{\epsilon})\boldsymbol{\epsilon}\|\mathbf{h}_\theta(\mathbf{x}_t, \sigma_t, y) - \mathbf{x}\|_2^2 d\boldsymbol{\epsilon}$$

$$=\frac{D}{\sigma_t}\max_{\|\mathbf{u}\|_2=1}\underbrace{\mathbf{u}^\top\int p(\boldsymbol{\epsilon})\boldsymbol{\epsilon}\frac{\|\mathbf{h}_\theta(\mathbf{x}_t, \sigma_t, y) - \mathbf{x}\|_2^2}{D}d\boldsymbol{\epsilon}}_{\text{The term in Salman et al. [38]}}$$

$$\leq\frac{D}{\sigma_t}\sqrt{\frac{2}{\pi}}.$$

The last inequality holds since $\frac{\|\mathbf{h}_\theta(\mathbf{x}_t, \sigma_t, y) - \mathbf{x}\|_2^2}{D}$ is a function in $[0, 1]^D$, thus it satisfies the condition of Lemma 1 in Salman et al. [38]. For the second term, since $\mathbf{x} \in [0, 1]^D$ and $\mathbf{h}_\theta(\mathbf{x}_t, \sigma_t, y) \in [0, 1]^D$, hence $\mathbf{h}_\theta(\mathbf{x}_t, \sigma_t, y) - \mathbf{x} \in [0, 1]^D$, consequently,

$$\|\mathbb{E}_{\boldsymbol{\epsilon}}[(\mathbf{h}_\theta(\mathbf{x}_t, \sigma_t, y) - \mathbf{x})\frac{2}{\sigma_t}]\|_2 = \|\mathbb{E}_{\boldsymbol{\epsilon}}[(\mathbf{h}_\theta(\mathbf{x}_t, \sigma_t, y) - \mathbf{x})]\|_2\frac{2}{\sigma_t} \leq \sqrt{D} \cdot \frac{2}{\sigma_t}.$$

Therefore,

$$\|\frac{d}{d\mathbf{x}}\mathbb{E}_{\boldsymbol{\epsilon}}[\|\mathbf{h}_\theta(\mathbf{x}_t, \sigma_t, y) - \mathbf{x}\|_2^2]\|_2 \leq \frac{D}{\sigma_t}\sqrt{\frac{2}{\pi}} + \sqrt{D} \cdot \frac{2}{\sigma_t} = \frac{1}{\sigma_t}(D\sqrt{\frac{2}{\pi}} + 2\sqrt{D}). \tag{17}$$

Both Chen et al. [3], Li et al. [26], Clark and Jaini [5] formulate the logit of the diffusion classifier as $\sum_{t=1}^T \mathbb{E}_{\boldsymbol{\epsilon}}[\|\mathbf{h}_\theta(\mathbf{x}_t, \sigma_t, y) - \mathbf{x}\|_2^2]$. However, in practice, both Chen et al. [3] and Li et al. [26] use MSE loss rather than L2 loss in the diffusion classifier. In other words, they multiply the logit by $\frac{1}{DT}$. Therefore, in this paper, we directly formulate the diffusion classifier using the MSE loss, as demonstrated in Eq. (6). Consequently, the maximum gradient norm of the logits in the diffusion classifier is:

$$\|\frac{d}{d\mathbf{x}}\frac{1}{DT}\sum_{t=1}^T w_t\mathbb{E}_{\boldsymbol{\epsilon}}[\|\mathbf{h}_\theta(\mathbf{x}_t, \sigma_t, y) - \mathbf{x}\|_2^2]\|_2 \leq \frac{1}{T}\sum_{t=1}^T\frac{w_t}{\sigma_t}\left(\sqrt{\frac{2}{\pi}} + \frac{2}{\sqrt{D}}\right). \tag{18}$$

*Remark* A.4. There may be concerns regarding how the selection of temperature (whether multiplying logits by $\frac{1}{DT}$) can significantly influence certified robustness. However, this is not the case. The temperature simultaneously affects both the scale of output from the diffusion classifier and the Lipschitz scale, with these scales changing proportionately. This is analogous to multiplying a function by a constant $k$: the Lipschitz constant will increase by a factor of $k$, but the gap between the outputs will also increase by the same factor, thus leaving the certified robustness unchanged.

Eq. (18) provides the maximum gradient norm (Lipschitz constant) in the logit space. One can already use Eq. (18) to perform certified robustness in the logit space, which is precisely what we did in our experiments. However, since our diffusion classifier is defined by the class probabilities rather

than the logits, to ensure consistency with our main text, we will derive the Lipschitz constant of the diffusion classifier in the class probabilities space in the following.

Let's define one-hot vector $\mathbf{e}_i$ in $\mathbb{R}^n$ to be the vector where the $i$-th element is 1 and all other elements are 0. Hinton et al. [10] proves that $\frac{\partial}{\partial \mathbf{x}} \log \mathrm{softmax}(\mathbf{x}/\beta)_y = \frac{1}{\beta}(\mathbf{e}_y - \mathrm{softmax}(\mathbf{x}/\beta))$, where $\beta$ is the softmax temperature. Consequently,

$$\frac{\partial}{\partial \mathbf{x}} \mathrm{softmax}(\mathbf{x}/\beta)_y = \left(\frac{\partial}{\partial \mathbf{x}} \log \mathrm{softmax}(\mathbf{x}/\beta)_y\right) \mathrm{softmax}(\mathbf{x}/\beta)_y$$

$$= (\mathbf{e}_y - \mathrm{softmax}(\mathbf{x}/\beta)) \frac{\mathrm{softmax}(\mathbf{x}/\beta)_y}{\beta}.$$

We can derive the maximum $\ell_2$ norm of the gradient as

$$\|\frac{\partial}{\partial \mathbf{x}_0} p(y|\mathbf{x}_0)\|_2$$

$$= \|\frac{\partial}{\partial \mathbf{x}_0} \mathrm{softmax}(f(\mathbf{x}))_y\|_2$$

$$= \|\sum_{y=1}^{K} \frac{\partial \mathrm{softmax}(f(\mathbf{x}))_y}{\partial(-\frac{1}{DT}\sum_{t=1}^{T} \mathbb{E}_{\boldsymbol{\epsilon}}\left[w_t\|\mathbf{h}_\theta(\mathbf{x}_t, t, y) - \mathbf{x}_0\|_2^2\right])} \frac{\partial(-\frac{1}{DT}\sum_{t=1}^{T} \mathbb{E}_{\boldsymbol{\epsilon}}\left[w_t\|\mathbf{h}_\theta(\mathbf{x}_t, t, y) - \mathbf{x}_0\|_2^2\right])}{\partial \mathbf{x}_0}\|_2$$

$$\leq \sum_{y=1}^{K} \|\frac{\partial \mathrm{softmax}(f(\mathbf{x}))_y}{\partial(-\frac{1}{DT}\sum_{t=1}^{T} \mathbb{E}_{\boldsymbol{\epsilon}}\left[w_t\|\mathbf{h}_\theta(\mathbf{x}_t, t, y) - \mathbf{x}_0\|_2^2\right])}\|_2 \|\frac{\partial(-\frac{1}{DT}\sum_{t=1}^{T} \mathbb{E}_{\boldsymbol{\epsilon}}\left[w_t\|\mathbf{h}_\theta(\mathbf{x}_t, t, y) - \mathbf{x}_0\|_2^2\right])}{\partial \mathbf{x}_0}\|_2$$

$$\leq \frac{1}{2\sqrt{2}} \sum_{i=1}^{T} \frac{w_i}{\sigma_i T}(\sqrt{\frac{2}{\pi}} + \frac{2}{\sqrt{D}}).$$

The last step is get by using Lemma A.2 for the first term and Eq. (18) for the second term. Denote $\mathbf{u}$ as a unit vector, the Lipschitz constant of the diffusion classifier $p(y|\mathbf{x})$ is

$$\max_{\mathbf{u}} \mathbf{u}^\top \frac{\partial}{\partial \mathbf{x}} p(y|\mathbf{x}) = \max_{\mathbf{u}} \|\mathbf{u}^\top \frac{\partial}{\partial \mathbf{x}} p(y|\mathbf{x})\|_2 = \|\frac{\partial}{\partial \mathbf{x}} p(y|\mathbf{x})\|_2 = \frac{1}{2\sqrt{2}} \sum_{i=1}^{T} \frac{w_i}{\sigma_i T}(\sqrt{\frac{2}{\pi}} + \frac{2}{\sqrt{D}}) \tag{19}$$

It is important to note that this represents the upper bound of the Lipschitz constant for the diffusion classifier. In practical scenarios, the actual Lipschitz constant is much smaller, due to the conservative nature of the inequalities used in the derivation. To gain an intuitive understanding of the practical Lipschitz constant for diffusion classifiers, we measure the gradient norm of the classifier on clean and noisy data. Furthermore, employing the algorithm from Chen et al. [4], we empirically determine the maximum gradient norm. Our results on the CIFAR-10 test set indicate that the gradient norm is less than 0.02, suggesting that the Lipschitz constant for our diffusion classifier is smaller than 0.02 on this dataset.

### A.3 Stronger Randomized Smoothing When $f$ Possess Lipschitzness

In this section, we will show that if $f$ has a smaller Lipschitz constant, it will induce a more smoothed function $g$, thus has a higher certified robustness. Here we discuss a simple case: *the weak law of randomized smoothing* proposed in Salman et al. [38] with $\sigma = 1$. We complement the derivation of this law with much more details, and we hope our detailed explanation could assist newcomers in the field in quickly grasping the key concepts in randomized smoothing.

To derive the Lipschitz constant of the smoothed function $g$, we only need to derive the maximum dot product between the gradient of $g$ and a unit vector $\mathbf{u}$ for the worst $f$.

$$\max_f \mathbf{u}^\top \nabla_{\mathbf{x}} g(\mathbf{x}) = \max_f \mathbf{u}^\top \nabla_{\mathbf{x}} \mathbb{E}_{\boldsymbol{\epsilon}}[f(\mathbf{x} + \boldsymbol{\epsilon})]$$

$$= \max_f \mathbf{u}^\top \nabla_{\mathbf{x}} \int p(\boldsymbol{\epsilon}) f(\mathbf{x} + \boldsymbol{\epsilon}) d\boldsymbol{\epsilon}$$

$$= \max_f \mathbf{u}^\top \nabla_{\mathbf{x}} \frac{1}{(2\pi)^{n/2}} \int \exp\left(-\frac{\|\boldsymbol{\epsilon}\|_2^2}{2}\right) f(\mathbf{x} + \boldsymbol{\epsilon}) d\boldsymbol{\epsilon}$$

$$= \max_f \mathbf{u}^\top \nabla_{\mathbf{x}} \frac{1}{(2\pi)^{n/2}} \int \exp\left(-\frac{\|\mathbf{t} - \mathbf{x}\|_2^2}{2}\right) f(\mathbf{t}) d\mathbf{t}$$

$$= \max_f \mathbf{u}^\top \frac{1}{(2\pi)^{n/2}} \int \exp\left(-\frac{\|\mathbf{t} - \mathbf{x}\|_2^2}{2}\right) (\mathbf{t} - \mathbf{x}) f(\mathbf{t}) d\mathbf{t}$$

$$= \max_f \mathbf{u}^\top \frac{1}{(2\pi)^{n/2}} \int \exp\left(-\frac{\|\mathbf{t} - \mathbf{x}\|_2^2}{2}\right) (\mathbf{t} - \mathbf{x}) f(\mathbf{t} - \mathbf{x}) d\mathbf{t}$$

$$= \max_f \frac{1}{(2\pi)^{n/2}} \int \exp\left(-\frac{\|\mathbf{t} - \mathbf{x}\|_2^2}{2}\right) [\mathbf{u}^\top (\mathbf{t} - \mathbf{x})] f(\mathbf{t} - \mathbf{x}) d\mathbf{t}$$

In the next step, we transition to another orthogonal coordinate system, denoted as $\mathbf{u}, \mathbf{u}_2, \cdots, \mathbf{u}_D$. This change is made with the assurance that the determinant of the Jacobian matrix, representing the transformation from the old coordinates to the new ones, equals 1. We then decompose the vector $\mathbf{t} - \mathbf{x}$ in the new coordinate system as follows:

$$\mathbf{t} - \mathbf{x} = a_1 \mathbf{u} + a_2 \mathbf{u}_2 + a_3 \mathbf{u}_3 + \cdots + a_D \mathbf{u}_D.$$

Therefore,

$$\max_f \frac{1}{(2\pi)^{n/2}} \int \exp\left(-\frac{\|\mathbf{t} - \mathbf{x}\|_2^2}{2}\right) [\mathbf{u}^\top (\mathbf{t} - \mathbf{x})] f(\mathbf{t} - \mathbf{x}) d\mathbf{t}$$

$$= \max_f \frac{1}{(2\pi)^{n/2}} \int \exp\left(-\frac{\|a_1 \mathbf{u} + a_2 \mathbf{u}_2 + a_3 \mathbf{u}_3 + \cdots + a_D \mathbf{u}_D\|_2^2}{2}\right) a_1 f(\mathbf{a}) d\mathbf{a}$$

$$= \max_f \frac{1}{(2\pi)^{n/2}} \int \exp\left(-\frac{a_1^2 + a_2^2 + a_3^2 + \cdots + a_D^2}{2}\right) a_1 f(\mathbf{a}) d\mathbf{a}$$

$$= \max_f \frac{1}{(2\pi)^{n/2}} \int_{-\infty}^{\infty} \exp\left(-\frac{a_1^2}{2}\right) a_1 \int_{-\infty}^{\infty} \exp\left(-\frac{a_2^2}{2}\right) \cdots \int_{-\infty}^{\infty} \exp\left(-\frac{a_D^2}{2}\right) f(\mathbf{a}) da_1 da_2 \cdots da_D$$

$$\leq \max_f \frac{1}{(2\pi)^{n/2}} \int_{-\infty}^{\infty} \exp\left(-\frac{a_1^2}{2}\right) a_1 f(a_1) da_1 \int_{-\infty}^{\infty} \exp\left(-\frac{a_2^2}{2}\right) da_2 \cdots \int_{-\infty}^{\infty} \exp\left(-\frac{a_D^2}{2}\right) da_D$$

$$= \max_f \frac{1}{(2\pi)^{1/2}} \int_{-\infty}^{\infty} \exp\left(-\frac{a_1^2}{2}\right) a_1 f(a_1) da_1$$

$$\leq \max_f \frac{1}{(2\pi)^{1/2}} \int_0^{\infty} \exp\left(-\frac{s^2}{2}\right) s \, ds.$$

We could get easily the result by change of variable:

$$\frac{1}{(2\pi)^{1/2}} \int_0^{\infty} \exp\left(-\frac{s^2}{2}\right) s \, ds = \frac{1}{(2\pi)^{1/2}} \left(-\exp\left(-\frac{s^2}{2}\right)\right)\Big|_0^{+\infty} = \frac{1}{\sqrt{2\pi}}.$$

The classifier will robustly classify the input data as long as the probability of classify the noisy data as the correct class is greater than the probability of classifying the noisy data as the wrong class. If one estimate a lower bound $\underline{p_A}$ of the accuracy of the correct class over noisy sample and a upper bound $\overline{p_B}$ of the probability of classify the noisy input as wrong class using the Clopper-Pearson interval, we could get the robust radius by

$$R = \sqrt{\frac{\pi}{2}} (\underline{p_A} - \overline{p_B}),$$

which will ensure that for any $\|\boldsymbol{\delta}\|_2 \leq R$ and any wrong class $\hat{y} \neq y$:

$$g(\mathbf{x} + \boldsymbol{\delta})_y = P(f(\mathbf{x} + \boldsymbol{\delta} + \boldsymbol{\epsilon}) = y) > g(\mathbf{x} + \boldsymbol{\delta})_{\hat{y}} = P(f(\mathbf{x} + \boldsymbol{\delta} + \boldsymbol{\epsilon}) = \hat{y}).$$

Hence, $g(x)$ will robustly classify the input data.

When $f$ satisfies the Lipschitz condition with Lipschitz constant $L$, since it is impossible to be one when $x > 0$ and zero when $x \leq 0$, the inequality here can be tighter, so we can get a smaller Lipschitz

constant. In fact, in this case, the maximum Lipschitz constant of $g$ is:

$$\max_f \frac{1}{\sqrt{2\pi}} \int_{-\infty}^{+\infty} \exp(-s^2/2) s f(s) ds$$
$$s.t., \ 0 \le f(s) \le 1, \ f(s) \text{ is } L\text{-lipschitz}$$

However, obtaining an analytical solution for the certified radius when $f$ adheres to Lipschitz continuity appears infeasible. To understand why, consider the following: firstly, $f$ should approach zero as $x$ tends toward negative infinity and one as $x$ tends toward positive infinity. Secondly, $f$ must be an increasing function, with only one interval of increase. Additionally, within this interval, $f$ is either zero or one outside the bounds of increase, and it must take a linear form within, with a slope of $\frac{1}{L}$. We define the left endpoint of this increasing interval as $a$, which necessarily lies in the range $[-\frac{1}{L}, 0]$. Also notice that both $a = -\frac{1}{L}$ and $a = 0$ yield the same certified radius. However, when $a \in (-\frac{1}{L}, 0)$, the certified radius must be smaller. To see why, let's consider

$$f_a(s) = \begin{cases} 0, & s \le a \\ L(s-a), & a \le s \le a + \frac{1}{L} \\ 1, & s \ge a + \frac{1}{L} \end{cases}, \text{ with special case } f_0(s) = \begin{cases} 0, & s \le 0 \\ Ls, & 0 \le s \le \frac{1}{L} \\ 1, & s \ge \frac{1}{L} \end{cases}.$$

Denote $\hat{f}_a(x) = \frac{1}{\sqrt{2\pi}} \exp(-x^2/2) x f_a(x)$. The difference of Lipschitz constant between two randomized function is

$$\int_{-\infty}^{+\infty} [\hat{f}_0(s) - \hat{f}_a(s)] ds = \int_a^0 [\hat{f}_0(s) - \hat{f}_a(s)] + \int_0^{-a} [\hat{f}_0(s) - \hat{f}_a(s)] + \int_{-a}^{1/L} [\hat{f}_0(s) - \hat{f}_a(s)]$$

$$= \int_0^{-a} [-aL - \hat{f}_a(s)] + \int_{-a}^{1/L} [\hat{f}_0(s) - \hat{f}_a(s)] \le 0$$

Consequently, the corner case for $a$ must lie within the open interval $(-1/L, 0)$. However, the corresponding integral lacks an analytical solution, and taking the derivative of it results in a function whose zero points are indeterminable. Therefore, obtaining an analytical solution for the improvement in certified robustness when a function exhibits Lipschitz continuity is challenging. Nevertheless, we can approximate the result with numerical algorithms, and it is evident that this Lipschitzness leads to a non-trivial enhancement in certified robustness of randomized smoothing.

### A.4 ELBO for Noisy Data in EPNDC

Similar to Ho et al. [11], we derive the ELBO as

$$\log p(\mathbf{x}_\tau) = \log \int \frac{p(\mathbf{x}_{\tau:T}) q(\mathbf{x}_{\tau+1:T}|\mathbf{x}_\tau)}{q(\mathbf{x}_{\tau+1:T}|\mathbf{x}_\tau)} d\mathbf{x}_{\tau+1:T}$$

$$= \log \mathbb{E}_{q(\mathbf{x}_{\tau+1:T}|\mathbf{x}_\tau)} \left[ \frac{p(\mathbf{x}_{\tau:T})}{q(\mathbf{x}_{\tau+1:T}|\mathbf{x}_\tau)} \right]$$

$$= \log \mathbb{E}_{q(\mathbf{x}_{\tau+1:T}|\mathbf{x}_\tau)} \left[ \frac{p(\mathbf{x}_T) p(\mathbf{x}_{\tau:T-1}|\mathbf{x}_T)}{q(\mathbf{x}_{\tau+1:T}|\mathbf{x}_\tau)} \right]$$

$$\ge \mathbb{E}_{q(\mathbf{x}_{\tau+1:T}|\mathbf{x}_\tau)} \left[ \log \frac{p(\mathbf{x}_T) p(\mathbf{x}_{\tau:T-1}|\mathbf{x}_T)}{q(\mathbf{x}_{\tau+1:T}|\mathbf{x}_\tau)} \right]$$

$$= \mathbb{E}_{q(\mathbf{x}_{\tau+1:T}|\mathbf{x}_\tau)} \left[ \log \frac{p(\mathbf{x}_T) \prod_{t=\tau}^{T-1} p(\mathbf{x}_t|\mathbf{x}_{t+1})}{\prod_{t=\tau}^{T-1} q(\mathbf{x}_{t+1}|\mathbf{x}_t, \mathbf{x}_\tau)} \right]$$

$$= \mathbb{E}_{q(\mathbf{x}_{\tau+1:T}|\mathbf{x}_\tau)} \left[ \log \frac{p(\mathbf{x}_T) \prod_{t=\tau}^{T-1} p(\mathbf{x}_t|\mathbf{x}_{t+1})}{\prod_{t=\tau}^{T-1} \frac{q(\mathbf{x}_{t+1}|\mathbf{x}_\tau) q(\mathbf{x}_t|\mathbf{x}_{t+1}, \mathbf{x}_\tau)}{q(\mathbf{x}_t|\mathbf{x}_\tau)}} \right]$$

$$= \mathbb{E}_{q(\mathbf{x}_{\tau+1:T}|\mathbf{x}_\tau)} \left[ \log \frac{p(\mathbf{x}_T) \prod_{t=\tau}^{T-1} p(\mathbf{x}_t|\mathbf{x}_{t+1})}{\prod_{t=\tau}^{T-1} q(\mathbf{x}_t|\mathbf{x}_{t+1}, \mathbf{x}_\tau)} - \log \prod_{t=\tau}^{T-1} \frac{q(\mathbf{x}_{t+1}|\mathbf{x}_\tau)}{q(\mathbf{x}_t|\mathbf{x}_\tau)} \right]$$

$$= \mathbb{E}_{q(\mathbf{x}_{\tau+1:T}|\mathbf{x}_\tau)}[\log \frac{p(\mathbf{x}_T)\prod_{t=\tau}^{T-1}p(\mathbf{x}_t|\mathbf{x}_{t+1})}{\prod_{t=\tau}^{T-1}q(\mathbf{x}_t|\mathbf{x}_{t+1},\mathbf{x}_\tau)} - \log \frac{q(\mathbf{x}_T|\mathbf{x}_\tau)}{q(\mathbf{x}_\tau|\mathbf{x}_\tau)}]$$

$$= \mathbb{E}_{q(\mathbf{x}_{\tau+1:T}|\mathbf{x}_\tau)}[\log \frac{\prod_{t=\tau}^{T-1}p(\mathbf{x}_t|\mathbf{x}_{t+1})}{\prod_{t=\tau}^{T-1}q(\mathbf{x}_t|\mathbf{x}_{t+1},\mathbf{x}_\tau)} - \log \frac{q(\mathbf{x}_T|\mathbf{x}_\tau)}{p(\mathbf{x}_T)}]$$

$$= \mathbb{E}_{q(\mathbf{x}_{\tau+1:T}|\mathbf{x}_\tau)}[\log \frac{\prod_{t=\tau}^{T-1}p(\mathbf{x}_t|\mathbf{x}_{t+1})}{\prod_{t=\tau}^{T-1}q(\mathbf{x}_t|\mathbf{x}_{t+1},\mathbf{x}_\tau)}] - \mathbb{E}_{q(\mathbf{x}_T|\mathbf{x}_\tau)}[D_{\mathrm{KL}}(q(\mathbf{x}_T|\mathbf{x}_\tau)||p(\mathbf{x}_T))]$$

$$= \sum_{t=\tau}^{T-1} \mathbb{E}_{q(\mathbf{x}_t,\mathbf{x}_{t+1}|\mathbf{x}_\tau)}[\log \frac{p(\mathbf{x}_t|\mathbf{x}_{t+1})}{q(\mathbf{x}_t|\mathbf{x}_{t+1},\mathbf{x}_\tau)}] - \mathbb{E}_{q(\mathbf{x}_T|\mathbf{x}_\tau)}[D_{\mathrm{KL}}(q(\mathbf{x}_T|\mathbf{x}_\tau)||p(\mathbf{x}_T))]$$

$$= \sum_{t=\tau}^{T-1} \mathbb{E}_{q(\mathbf{x}_{t+1}|\mathbf{x}_\tau),q(\mathbf{x}_t|\mathbf{x}_{t+1},\mathbf{x}_\tau)}[\log \frac{p(\mathbf{x}_t|\mathbf{x}_{t+1})}{q(\mathbf{x}_t|\mathbf{x}_{t+1},\mathbf{x}_\tau)}] - \mathbb{E}_{q(\mathbf{x}_T|\mathbf{x}_\tau)}[D_{\mathrm{KL}}(q(\mathbf{x}_T|\mathbf{x}_\tau)||p(\mathbf{x}_T))]$$

$$= \log p(\mathbf{x}_\tau|\mathbf{x}_{\tau+1}) - \sum_{t=\tau+1}^{T-1} \mathbb{E}_{q(\mathbf{x}_{t+1}|\mathbf{x}_\tau)}[D_{\mathrm{KL}}(q(\mathbf{x}_t|\mathbf{x}_{t+1},\mathbf{x}_\tau)||p(\mathbf{x}_t|\mathbf{x}_{t+1}))]$$

$$- \mathbb{E}_{q(\mathbf{x}_T|\mathbf{x}_\tau)}[D_{\mathrm{KL}}(q(\mathbf{x}_T|\mathbf{x}_\tau)||p(\mathbf{x}_T))]$$

$$= -\sum_{t=\tau+1}^{T-1} \mathbb{E}_{q(\mathbf{x}_{t+1}|\mathbf{x}_\tau)}[D_{\mathrm{KL}}(q(\mathbf{x}_t|\mathbf{x}_{t+1},\mathbf{x}_\tau)||p(\mathbf{x}_t|\mathbf{x}_{t+1}))] + C$$

This could be understood as a generalization of the ELBO in Ho et al. [11], which is a special case when $\tau = 0$. Notice that the summation of KL divergence in ELBO of $\log p(\mathbf{x}_\tau)$ start from $\tau + 1$ and end at $T$, while that of $\log p(\mathbf{x}_0)$ start from 1. Besides, the posterior is $q(\mathbf{x}_t|\mathbf{x}_{t+1},\mathbf{x}_\tau)$ instead of $q(\mathbf{x}_t|\mathbf{x}_{t+1},\mathbf{x}_0)$.

Similarly, the conditional ELBO is give by:

$$\log p(\mathbf{x}_\tau|y) \geq \log p(\mathbf{x}_\tau|\mathbf{x}_{\tau+1},y) - \mathbb{E}_{q(\mathbf{x}_T|\mathbf{x}_\tau)}[D_{\mathrm{KL}}(q(\mathbf{x}_T|\mathbf{x}_\tau)||p(\mathbf{x}_T))]$$

$$- \sum_{t=\tau+1}^{T-1} \mathbb{E}_{q(\mathbf{x}_{t+1}|\mathbf{x}_\tau)}[D_{\mathrm{KL}}(q(\mathbf{x}_t|\mathbf{x}_{t+1},\mathbf{x}_\tau,y)||p(\mathbf{x}_t|\mathbf{x}_{t+1},y))]$$

$$= -\sum_{t=\tau+1}^{T-1} \mathbb{E}_{q(\mathbf{x}_{t+1}|\mathbf{x}_\tau)}[D_{\mathrm{KL}}(q(\mathbf{x}_t|\mathbf{x}_{t+1},\mathbf{x}_\tau,y)||p(\mathbf{x}_t|\mathbf{x}_{t+1},y))] + C_5.$$

This is equivalent to adding the condition $y$ to all posterior distributions in the unconditional ELBO.

### A.5 The Analytical Form of the KL Divergence in ELBO

The KL divergence in Theorem 3.2 is the expectation of the log ratio between posterior and predicted reverse distribution, which requires integrating over the entire space. To compute this KL divergence more efficiently, we derives its analytical form:

$$q(\mathbf{x}_t|\mathbf{x}_{t+1},\mathbf{x}_\tau) = \frac{q(\mathbf{x}_t|\mathbf{x}_\tau)q(\mathbf{x}_{t+1}|\mathbf{x}_t)}{q(\mathbf{x}_{t+1}|\mathbf{x}_\tau)}$$

$$= \frac{\mathcal{N}(\mathbf{x}_t|\mathbf{x}_\tau,(\sigma_t^2-\sigma_\tau^2)I)\mathcal{N}(\mathbf{x}_{t+1}|\mathbf{x}_t,(\sigma_{t+1}^2-\sigma_t^2)I)}{\mathcal{N}(\mathbf{x}_{t+1}|\mathbf{x}_\tau,(\sigma_{t+1}^2-\sigma_\tau^2)I)}$$

$$= \frac{\frac{1}{(2\pi(\sigma_t^2-\sigma_\tau^2))^{n/2}}\exp(\frac{\|\mathbf{x}_t-\mathbf{x}_\tau\|^2}{-2(\sigma_t^2-\sigma_\tau^2)})\frac{1}{(2\pi(\sigma_{t+1}^2-\sigma_t^2))^{n/2}}\exp(\frac{\|\mathbf{x}_{t+1}-\mathbf{x}_t\|^2}{-2(\sigma_{t+1}^2-\sigma_t^2)})}{\frac{1}{(2\pi(\sigma_{t+1}^2-\sigma_\tau^2))^{n/2}}\exp(\frac{\|\mathbf{x}_{t+1}-\mathbf{x}_\tau\|^2}{-2(\sigma_{t+1}^2-\sigma_\tau^2)})}.$$

Since the likelihood distribution $q(\mathbf{x}_{t+1}|\mathbf{x}_t)$ and the prior distribution $q(\mathbf{x}_t|\mathbf{x}_\tau)$ are both Gaussian distribution, the posterior $q(\mathbf{x}_t|\mathbf{x}_{t+1},\mathbf{x}_\tau)$ is also a Gaussian distribution. Therefore, we only need to derive the expectation and the covariance matrix. In the following, instead of derive the $q(\mathbf{x}_t|\mathbf{x}_{t+1},\mathbf{x}_\tau)$ using equations, we use some trick to simplify the derivation:

$$q(\mathbf{x}_t|\mathbf{x}_{t+1},\mathbf{x}_\tau)$$

$$\propto \exp\left(\frac{\|\mathbf{x}_t - \mathbf{x}_\tau\|^2}{-2(\sigma_t^2 - \sigma_\tau^2)} + \frac{\|\mathbf{x}_{t+1} - \mathbf{x}_t\|^2}{-2(\sigma_{t+1}^2 - \sigma_t^2)} - \frac{\|\mathbf{x}_{t+1} - \mathbf{x}_\tau\|^2}{-2(\sigma_{t+1}^2 - \sigma_\tau^2)}\right)$$

$$= \exp\left(-\frac{1}{2}\left[\frac{\|\mathbf{x}_t - \mathbf{x}_\tau\|^2}{(\sigma_t^2 - \sigma_\tau^2)} + \frac{\|\mathbf{x}_{t+1} - \mathbf{x}_t\|^2}{(\sigma_{t+1}^2 - \sigma_t^2)} - \frac{\|\mathbf{x}_{t+1} - \mathbf{x}_\tau\|^2}{(\sigma_{t+1}^2 - \sigma_\tau^2)}\right]\right)$$

$$= \exp\left(-\frac{1}{2}\left[\frac{\|\mathbf{x}_t\|^2 - 2\mathbf{x}_t^T \mathbf{x}_\tau}{(\sigma_t^2 - \sigma_\tau^2)} + \frac{\|\mathbf{x}_t\|^2 - 2\mathbf{x}_{t+1}^T \mathbf{x}_t}{(\sigma_{t+1}^2 - \sigma_t^2)} + C(\mathbf{x}_{t+1}, \mathbf{x}_\tau)\right]\right)$$

$$= \exp\left(-\frac{1}{2}\left[\frac{\|\mathbf{x}_t\|^2}{(\sigma_t^2 - \sigma_\tau^2)} - \frac{2\mathbf{x}_t^T \mathbf{x}_\tau}{(\sigma_t^2 - \sigma_\tau^2)} + \frac{\|\mathbf{x}_t\|^2}{(\sigma_{t+1}^2 - \sigma_t^2)} - \frac{2\mathbf{x}_{t+1}^T \mathbf{x}_t}{(\sigma_{t+1}^2 - \sigma_t^2)} + C(\mathbf{x}_{t+1}, \mathbf{x}_\tau)\right]\right)$$

$$= \exp\left(-\frac{1}{2}\left[\left(\frac{1}{(\sigma_t^2 - \sigma_\tau^2)} + \frac{1}{(\sigma_{t+1}^2 - \sigma_t^2)}\right)\|\mathbf{x}_t\|^2 - 2\left(\frac{\mathbf{x}_\tau^T}{(\sigma_t^2 - \sigma_\tau^2)} + \frac{\mathbf{x}_{t+1}^T}{(\sigma_{t+1}^2 - \sigma_t^2)}\right)\mathbf{x}_t + C(\mathbf{x}_{t+1}, \mathbf{x}_\tau)\right]\right)$$

$$= \exp\left(-\frac{1}{2}\left[\frac{(\sigma_t^2 - \sigma_\tau^2) + (\sigma_{t+1}^2 - \sigma_t^2)}{(\sigma_t^2 - \sigma_\tau^2)(\sigma_{t+1}^2 - \sigma_t^2)}\|\mathbf{x}_t\|^2 - 2\left(\frac{\mathbf{x}_\tau^T}{(\sigma_t^2 - \sigma_\tau^2)} + \frac{\mathbf{x}_{t+1}^T}{(\sigma_{t+1}^2 - \sigma_t^2)}\right)\mathbf{x}_t + C(\mathbf{x}_{t+1}, \mathbf{x}_\tau)\right]\right)$$

$$= \exp\left(-\frac{1}{2}\left[\frac{\sigma_{t+1}^2 - \sigma_\tau^2}{(\sigma_t^2 - \sigma_\tau^2)(\sigma_{t+1}^2 - \sigma_t^2)}\|\mathbf{x}_t\|^2 - 2\left(\frac{\mathbf{x}_\tau^T}{(\sigma_t^2 - \sigma_\tau^2)} + \frac{\mathbf{x}_{t+1}^T}{(\sigma_{t+1}^2 - \sigma_t^2)}\right)\mathbf{x}_t + C(\mathbf{x}_{t+1}, \mathbf{x}_\tau)\right]\right)$$

$$= \exp\left(-\frac{1}{2}\left[\frac{\sigma_{t+1}^2 - \sigma_\tau^2}{(\sigma_t^2 - \sigma_\tau^2)(\sigma_{t+1}^2 - \sigma_t^2)}\left(\|\mathbf{x}_t\|^2 - 2\frac{\frac{\mathbf{x}_\tau^T}{(\sigma_t^2 - \sigma_\tau^2)} + \frac{\mathbf{x}_{t+1}^T}{(\sigma_{t+1}^2 - \sigma_t^2)}}{\frac{\sigma_{t+1}^2 - \sigma_\tau^2}{(\sigma_t^2 - \sigma_\tau^2)(\sigma_{t+1}^2 - \sigma_t^2)}}\mathbf{x}_t\right) + C(\mathbf{x}_{t+1}, \mathbf{x}_\tau)\right]\right)$$

$$= \exp\left(-\frac{1}{2}\left[\frac{\sigma_{t+1}^2 - \sigma_\tau^2}{(\sigma_t^2 - \sigma_\tau^2)(\sigma_{t+1}^2 - \sigma_t^2)}\left(\|\mathbf{x}_t\|^2 - 2\frac{(\sigma_{t+1}^2 - \sigma_t^2)\mathbf{x}_\tau^T + (\sigma_t^2 - \sigma_\tau^2)\mathbf{x}_{t+1}^T}{\sigma_{t+1}^2 - \sigma_\tau^2}\mathbf{x}_t\right) + C(\mathbf{x}_{t+1}, \mathbf{x}_\tau)\right]\right)$$

$$= \exp\left(-\frac{1}{2\frac{(\sigma_t^2 - \sigma_\tau^2)(\sigma_{t+1}^2 - \sigma_t^2)}{\sigma_{t+1}^2 - \sigma_\tau^2}}\left[\left(\mathbf{x}_t - \frac{(\sigma_{t+1}^2 - \sigma_t^2)\mathbf{x}_\tau + (\sigma_t^2 - \sigma_\tau^2)\mathbf{x}_{t+1}}{\sigma_{t+1}^2 - \sigma_\tau^2}\right)^2 + C(\mathbf{x}_{t+1}, \mathbf{x}_\tau)\right]\right)$$

$$\propto \mathcal{N}\left(\mathbf{x}_t; \frac{(\sigma_{t+1}^2 - \sigma_t^2)\mathbf{x}_\tau + (\sigma_t^2 - \sigma_\tau^2)\mathbf{x}_{t+1}}{\sigma_{t+1}^2 - \sigma_\tau^2}, \frac{(\sigma_t^2 - \sigma_\tau^2)(\sigma_{t+1}^2 - \sigma_t^2)}{\sigma_{t+1}^2 - \sigma_\tau^2}\mathbf{I}\right).$$

The KL divergence between the posterior and predicted reverse distribution is

$$D_{\mathrm{KL}}(q(\mathbf{x}_t|\mathbf{x}_{t+1}, \mathbf{x}_\tau)\|p_\theta(\mathbf{x}_t|\mathbf{x}_{t+1}))$$

$$= D_{\mathrm{KL}}\left(\mathcal{N}\left(\mathbf{x}_t; \frac{(\sigma_{t+1}^2 - \sigma_t^2)\mathbf{x}_\tau + (\sigma_t^2 - \sigma_\tau^2)\mathbf{x}_{t+1}}{\sigma_{t+1}^2 - \sigma_\tau^2}, \frac{(\sigma_t^2 - \sigma_\tau^2)(\sigma_{t+1}^2 - \sigma_t^2)}{\sigma_{t+1}^2 - \sigma_\tau^2}\mathbf{I}\right)\right\|$$

$$\mathcal{N}\left(\mathbf{x}_t; \frac{(\sigma_{t+1}^2 - \sigma_t^2)h(\mathbf{x}_{t+1}, \sigma_{t+1}) + \sigma_t^2 \mathbf{x}_{t+1}}{\sigma_{t+1}^2}, \tilde{\sigma}_t^2 \mathbf{I}\right)\right)$$

$$= \frac{1}{2\frac{(\sigma_t^2 - \sigma_\tau^2)(\sigma_{t+1}^2 - \sigma_t^2)}{\sigma_{t+1}^2 - \sigma_\tau^2}}\|\mathbb{E}_{q(\mathbf{x}_t|\mathbf{x}_{t+1}, \mathbf{x}_\tau)}[\mathbf{x}_t] - \mathbb{E}_{p(\mathbf{x}_t|\mathbf{x}_{t+1})}[\mathbf{x}_t]\|^2 + C_2$$

$$= \frac{\sigma_{t+1}^2 - \sigma_\tau^2}{2(\sigma_t^2 - \sigma_\tau^2)(\sigma_{t+1}^2 - \sigma_t^2)}\|\mathbb{E}_{q(\mathbf{x}_t|\mathbf{x}_{t+1}, \mathbf{x}_\tau)}[\mathbf{x}_t] - \mathbb{E}_{p(\mathbf{x}_t|\mathbf{x}_{t+1})}[\mathbf{x}_t]\|^2 + C_2.$$

When $\tau = 0$, the result degrade to:

$$q(\mathbf{x}_t|\mathbf{x}_{t+1}, \mathbf{x}_0) = \mathcal{N}\left(\mathbf{x}_t; \frac{(\sigma_{t+1}^2 - \sigma_t^2)\mathbf{x}_0 + \sigma_t^2 \mathbf{x}_{t+1}}{\sigma_{t+1}^2}, \frac{\sigma_t^2(\sigma_{t+1}^2 - \sigma_t^2)}{\sigma_{t+1}^2}\mathbf{I}\right).$$

When $d\sigma_t := \sigma_{t+1} - \sigma_t \to 0$, the KL divergence between posterior and model prediction is simplified by:

$$\lim_{d\sigma_t \to 0} D_{\mathrm{KL}}(q(\mathbf{x}_t|\mathbf{x}_{t+1}, \mathbf{x}_0)\|p_\theta(\mathbf{x}_t|\mathbf{x}_{t+1}))$$

$$= \lim_{d\sigma_t \to 0} D_{\mathrm{KL}}\left(\mathcal{N}\left(\mathbf{x}_t; \frac{(\sigma_{t+1}^2 - \sigma_t^2)\mathbf{x}_0 + \sigma_t^2 \mathbf{x}_{t+1}}{\sigma_{t+1}^2}, \frac{\sigma_t^2(\sigma_{t+1}^2 - \sigma_t^2)}{\sigma_{t+1}^2}\mathbf{I}\right)\right\|$$

$$\mathcal{N}(\mathbf{x}_t; \frac{(\sigma_{t+1}^2 - \sigma_t^2)h(\mathbf{x}_{t+1}, \sigma_{t+1}) + \sigma_t^2 \mathbf{x}_{t+1}}{\sigma_{t+1}^2}, \tilde{\sigma}_t))$$

$$= \lim_{d\sigma_t \to 0} \frac{1}{2\frac{\sigma_t^2(\sigma_{t+1}^2 - \sigma_t^2)}{\sigma_{t+1}^2}} \| \frac{(\sigma_{t+1}^2 - \sigma_t^2)\mathbf{x}_0}{\sigma_{t+1}^2} - \frac{(\sigma_{t+1}^2 - \sigma_t^2)h(\mathbf{x}_{t+1}, \sigma_{t+1})}{\sigma_{t+1}^2} \|^2$$

$$= \lim_{d\sigma_t \to 0} \frac{(\sigma_{t+1}^2 - \sigma_t^2)^2}{2\frac{\sigma_t^2(\sigma_{t+1}^2 - \sigma_t^2)}{\sigma_{t+1}^2} \sigma_{t+1}^4} \|\mathbf{x}_0 - h(\mathbf{x}_{t+1}, \sigma_{t+1})\|^2$$

$$= \lim_{d\sigma_t \to 0} \frac{(\sigma_{t+1}^2 - \sigma_t^2)}{2\frac{\sigma_t^2}{\sigma_{t+1}^2}\sigma_{t+1}^4} \|\mathbf{x}_0 - h(\mathbf{x}_{t+1}, \sigma_{t+1})\|^2$$

$$= \lim_{d\sigma_t \to 0} \frac{(\sigma_{t+1}^2 - \sigma_t^2)}{2\sigma_{t+1}^4} \|\mathbf{x}_0 - h(\mathbf{x}_{t+1}, \sigma_{t+1})\|^2$$

$$= \frac{d\sigma_t}{\sigma_{t+1}^3} \|x_0 - h(\mathbf{x}_{t+1}, \sigma_{t+1})\|^2$$

$$= w(i)\|\mathbf{x}_0 - h(\mathbf{x}_{t+1}, \sigma_{t+1})\|^2,$$

which is consistent to the results in Kingma et al. [17] and Karras et al. [16].

### A.6 Deriving the Weight in EPNDC

If we interpret the shift in weight from $\frac{\sigma_{t+1} - \sigma_t}{\sigma_{t+1}}$ to $\frac{\sigma_t^2 + \sigma_d^2}{\sigma_t^2 \sigma_d^2} \frac{1}{\sqrt{2\pi}k_\sigma} \exp(-\frac{\|\log \sigma_t - k_\mu\|^2}{2k_\sigma^2})$ as a reweighting of $D_{\mathrm{KL}}(q(\mathbf{x}_t|\mathbf{x}_{t+1}, \mathbf{x}_0)\|p_\theta(\mathbf{x}_t|\mathbf{x}_{t+1}))$ by $\frac{\hat{w}_t}{w_t}$, a similar methodology can be applied to derive the weight for $D_{\mathrm{KL}}(q(\mathbf{x}_t|\mathbf{x}_{t+1}, \mathbf{x}_\tau)\|p_\theta(\mathbf{x}_t|\mathbf{x}_{t+1}))$:

$$\lim_{d\sigma_t \to 0} \frac{\hat{w}_t}{w_t} D_{\mathrm{KL}}(q(\mathbf{x}_t|\mathbf{x}_{t+1}, \mathbf{x}_\tau)\|p_\theta(\mathbf{x}_t|\mathbf{x}_{t+1}))$$

$$= \lim_{d\sigma_t \to 0} \frac{\hat{w}_t}{w_t} \frac{1}{2\frac{(\sigma_t^2 - \sigma_\tau^2)(\sigma_{t+1}^2 - \sigma_t^2)}{\sigma_{t+1}^2 - \sigma_\tau^2}} \|\mathbb{E}_{q(\mathbf{x}_t|\mathbf{x}_{t+1}, \mathbf{x}_\tau)}[\mathbf{x}_t] - \mathbb{E}_{p(\mathbf{x}_t|\mathbf{x}_{t+1})}[\mathbf{x}_t]\|_2^2$$

$$= \lim_{d\sigma_t \to 0} \frac{\hat{w}_t}{\frac{d\sigma_t}{\sigma_{t+1}^3}} \frac{1}{2\frac{(\sigma_t^2 - \sigma_\tau^2)(2\sigma_t d\sigma_t)}{\sigma_{t+1}^2 - \sigma_\tau^2}} \|\mathbb{E}_{q(\mathbf{x}_t|\mathbf{x}_{t+1}, \mathbf{x}_\tau)}[\mathbf{x}_t] - \mathbb{E}_{p(\mathbf{x}_t|\mathbf{x}_{t+1})}[\mathbf{x}_t]\|_2^2$$

$$= \lim_{d\sigma_t \to 0} \frac{\hat{w}_t}{\frac{(d\sigma_t)^2}{\sigma_{t+1}^2}} \frac{1}{4\frac{(\sigma_t^2 - \sigma_\tau^2)}{\sigma_{t+1}^2 - \sigma_\tau^2}} \|\mathbb{E}_{q(\mathbf{x}_t|\mathbf{x}_{t+1}, \mathbf{x}_\tau)}[\mathbf{x}_t] - \mathbb{E}_{p(\mathbf{x}_t|\mathbf{x}_{t+1})}[\mathbf{x}_t]\|_2^2$$

$$= \lim_{d\sigma_t \to 0} \frac{\hat{w}_t \sigma_{t+1}^2 (\sigma_{t+1}^2 - \sigma_\tau^2)}{4(\sigma_t^2 - \sigma_\tau^2)(d\sigma_t)^2} \|\mathbb{E}_{q(\mathbf{x}_t|\mathbf{x}_{t+1}, \mathbf{x}_\tau)}[\mathbf{x}_t] - \mathbb{E}_{p(\mathbf{x}_t|\mathbf{x}_{t+1})}[\mathbf{x}_t]\|_2^2$$

$$= \lim_{d\sigma_t \to 0} \frac{\sigma_t^2 + \sigma_d^2}{\sigma_t^2 \sigma_d^2} \frac{1}{\sqrt{2\pi}k_\sigma} \exp(-\frac{\|\log \sigma_t - k_\mu\|^2}{2k_\sigma^2}) \frac{\sigma_{t+1}^2(\sigma_{t+1}^2 - \sigma_\tau^2)}{4(\sigma_t^2 - \sigma_\tau^2)(d\sigma_t)^2}$$
$$\cdot \|\mathbb{E}_{q(\mathbf{x}_t|\mathbf{x}_{t+1}, \mathbf{x}_\tau)}[\mathbf{x}_t] - \mathbb{E}_{p(\mathbf{x}_t|\mathbf{x}_{t+1})}[\mathbf{x}_t]\|_2^2$$

$$= \lim_{d\sigma_t \to 0} \frac{\sigma_t^2 + \sigma_d^2}{\sigma_d^2} \frac{1}{\sqrt{2\pi}k_\sigma} \exp(-\frac{\|\log \sigma_t - k_\mu\|^2}{2k_\sigma^2}) \frac{\sigma_{t+1}^2 - \sigma_\tau^2}{4(\sigma_t^2 - \sigma_\tau^2)(\sigma_{t+1} - \sigma_t)^2}$$
$$\cdot \|\mathbb{E}_{q(\mathbf{x}_t|\mathbf{x}_{t+1}, \mathbf{x}_\tau)}[\mathbf{x}_t] - \mathbb{E}_{p(\mathbf{x}_t|\mathbf{x}_{t+1})}[\mathbf{x}_t]\|_2^2$$

We compare different weights in EPNDC. As demonstrated in Table 3, these weights are not as effective as the derived weight. Interestingly, simply setting the weight to zero when the standard deviation falls below a certain threshold significantly boosts the performance of the derived weight. This suggests that the latter part of the derived weight might be close to the optimal weight. We observe a similar phenomenon with the final weight we use, suggesting that the current weight might not be optimal. We defer the investigation into how to find the optimal weight, what constitutes this optimal weight, and why it is considered optimal to future work.

Table 3: The accuracy of EPNDC using the EDM checkpoint [16] on CIFAR-10 test set with various weight when $\sigma = 0.25$. Result are tested on the same subset with 512 images as we clarified in Sec. 4.1.

| Weight Name | Weight | Accuracy(%) |
|---|---|---|
| Normalized Weight | $\frac{1}{\|\mathbb{E}_{q(\mathbf{x}_t\|\mathbf{x}_{t+1},\mathbf{x}_\tau)}[\mathbf{x}_t] - \mathbb{E}_{p(\mathbf{x}_t\|\mathbf{x}_{t+1})}[\mathbf{x}_t]\|_2^2}$ | 81.6 |
| Derived Weight | $w_t$ | 67.5 |
| Truncated Derived Weight | $w_t \cdot \mathbb{I}\{\sigma_t > 1\}$ | 82.8 |
| Linear Weight | $\frac{w_T - w_0}{\sigma_t - \sigma_0} + w_0$ | 43.8 |
| Truncated Linear Weight | $(\frac{w_T - w_0}{\sigma_t - \sigma_0} + w_0) \cdot \mathbb{I}\{\sigma_t > 0.5\}$ | 77.0 |

We also attempt to learn an optimal weight for our EPNDC. However, the learning process is difficult due to the high variance of $\|\mathbb{E}_{q(\mathbf{x}_t|\mathbf{x}_{t+1},\mathbf{x}_\tau)}[\mathbf{x}_t] - \mathbb{E}_{p(\mathbf{x}_t|\mathbf{x}_{t+1})}[\mathbf{x}_t]\|_2^2$. Attempts to simplify the learning process through cubic spline interpolation for reducing the number of learned parameters are also time-consuming. Our work primarily focuses on robust classification of input data using a single off-the-shelf diffusion model. Therefore, we leave the exploration of this aspect for future research.

### A.7 ELBO for Noisy Data in APNDC

In this section, we show that APNDC is actually use the expectation of the ELBOs calculated from samples that are first denoised from the input and then have noise added back. In other word, APNDC approximate $\log p(\hat{\mathbf{x}}_\tau)$ by the lower bound of $\mathbb{E}_{p(\mathbf{x}_0|\hat{\mathbf{x}}_\tau)}\mathbb{E}_{q(\mathbf{x}_\tau|\mathbf{x}_0)}[\log p_\tau(\mathbf{x}_\tau)]$.

We first derive the lower bound for $\mathbb{E}_{q(\mathbf{x}_t|\mathbf{x}_0)}[\log p_\tau(\mathbf{x}_\tau)]$. The derivation of this lower bound is just the discrete case of the proof in Kingma and Gao [18]. We include it here only for completeness.

$$
\begin{aligned}
&\mathbb{E}_{q(\mathbf{x}_\tau|\mathbf{x}_0)}[\log p_\tau(\mathbf{x}_\tau)] \\
=&\int q(\mathbf{x}_\tau|\mathbf{x}_0)\frac{\log q(\mathbf{x}_\tau|\mathbf{x}_0)\log p_\tau(\mathbf{x}_\tau)}{\log q(\mathbf{x}_\tau|\mathbf{x}_0)} \\
=&\mathbb{E}_{q(\mathbf{x}_\tau|\mathbf{x}_0)}[\log q(\mathbf{x}_\tau|\mathbf{x}_0)] - D_{\mathrm{KL}}(q(\mathbf{x}_\tau|\mathbf{x}_0)\|p_\tau(\mathbf{x}_\tau)) \\
\geq&\mathbb{E}_{q(\mathbf{x}_\tau|\mathbf{x}_0)}[\log q(\mathbf{x}_\tau|\mathbf{x}_0)] - D_{\mathrm{KL}}(q(\mathbf{x}_{\tau:T}|\mathbf{x}_0)\|p(\mathbf{x}_{\tau:T})) \\
=&\mathbb{E}_{q(\mathbf{x}_\tau|\mathbf{x}_0)}[\log q(\mathbf{x}_\tau|\mathbf{x}_0)] - \sum_{t=\tau}^{T-1}[D_{\mathrm{KL}}(q(\mathbf{x}_{t:T}|\mathbf{x}_0)\|p(\mathbf{x}_{t:T})) \\
&- D_{\mathrm{KL}}(q(\mathbf{x}_{t+1:T}|\mathbf{x}_0)\|p(\mathbf{x}_{t+1:T}))] + D_{\mathrm{KL}}(q(\mathbf{x}_T|\mathbf{x}_0)\|p(\mathbf{x}_T)) \\
=&C_4 - \sum_{t=\tau}^{T-1}[D_{\mathrm{KL}}(q(\mathbf{x}_{t:T}|\mathbf{x}_0)\|p(\mathbf{x}_{t:T})) - D_{\mathrm{KL}}(q(\mathbf{x}_{t+1:T}|\mathbf{x}_0)\|p(\mathbf{x}_{t+1:T}))] \\
=&C_4 - \sum_{t=\tau}^{T-1}D_{\mathrm{KL}}(q(\mathbf{x}_t|\mathbf{x}_{t+1},\mathbf{x}_0)\|p(\mathbf{x}_t|\mathbf{x}_{t+1})) \\
=&C_4 - \sum_{t=\tau+1}^{T-1}w_\tau\mathbb{E}_{\mathbf{x}_t\sim q(\mathbf{x}_t|\mathbf{x}_0)}[\|\mathbf{h}_\theta(\mathbf{x}_t,\sigma_t) - \mathbf{x}\|_2^2].
\end{aligned}
$$

Given a input image $\mathbf{x}_\tau$, we use the ELBOs of $\mathbb{E}_{q(\hat{\mathbf{x}}_\tau|\mathbf{x}_0=\mathbf{h}_\theta(\mathbf{x}_\tau,\sigma_\tau))}[\log p_\tau(\hat{\mathbf{x}}_\tau)]$, rather than its own ELBO, to approximate its log likelihood:

$$
\log p(\mathbf{x}_\tau) \approx \mathbb{E}_{q(\hat{\mathbf{x}}_\tau|\mathbf{x}_0=\mathbf{h}_\theta(\mathbf{x}_\tau,\sigma_\tau))}[\log p_\tau(\hat{\mathbf{x}}_\tau)] \geq C_4 - \sum_{t=\tau+1}^{T-1}w_t\mathbb{E}_{\mathbf{x}_t\sim q(\mathbf{x}_t|\mathbf{x}_0)}[\|\mathbf{h}_\theta(\mathbf{x}_t,\sigma_t) - \mathbf{x}\|_2^2].
$$

Hence, APNDC is actually use the expectation of the ELBOs calculated from samples that are first denoised from the input and then have noise added back.

---

**Algorithm 3** Linear to EDM

---

1: **Require:** A pre-trained EDM $\mathbf{h}_\theta$, a noisy input image $\mathbf{x}_t$, noise level $t$, linear schedule $\{\alpha_i\}_{i=1}^T$ and $\{\sigma_i\}_{i=1}^T$.
2: Calculate the denoised image $\mathbf{x}_0$ using $\mathbf{h}_\theta$: $\mathbf{x}_0 = \mathbf{h}_\theta\left(\frac{\mathbf{x}_t}{\alpha_t}, \frac{\sigma_t}{\alpha_t}\right)$
3: **if** performing $\mathbf{x}_0$-prediction **then**
4:     **Return:** $\mathbf{x}_0$.
5: **end if**
6: Calculate the noise component $\epsilon$: $\epsilon = \frac{\mathbf{x}_t - \alpha_t \mathbf{x}_0}{\sigma_t}$
7: **if** performing $\epsilon$-prediction **then**
8:     **Return:** $\epsilon$.
9: **end if**

---

---

**Algorithm 4** EDM to Linear

---

1: **Require:** A pre-trained predictor $\mathbf{h}_\theta$, a noisy input image $\mathbf{x}_t$, noise level $\sigma$, linear schedule $\{\alpha_i\}_{i=1}^T$ and $\{\sigma_i\}_{i=1}^T$.
2: Calculate $t = \arg\min_t |\frac{\sigma_t}{\alpha_t} - \sigma|$;
3: **if** performing $\epsilon$-prediction **then**
4:     Predict $\epsilon = \epsilon(\alpha_t \mathbf{x}_t, t)$
5:     Calculate $\mathbf{x}_0 = \mathbf{x}_t - \sigma \cdot \epsilon$
6: **end if**
7: **if** performing $\mathbf{x}_0$-prediction **then**
8:     Predict $\mathbf{x}_0 = \epsilon(\alpha_t \mathbf{x}_t, t)$
9: **end if**
10: **Return:** $\mathbf{x}_0$.

---

### A.8 Converting Other Diffusion Models into Our Definition.

In this paper, we introduce a new definition for diffusion models, specifically as the discrete version of Karras et al. [16]. This definition encompasses various diffusions, including VE-SDE, VP-SDE, and methods like x-prediction, v-prediction, and epsilon-prediction, transforming their differences into the difference of parametrization in $\mathbf{h}_\theta(\cdot, \cdot)$, makes theoretical analysis extremely convenient. This operation also decouple the training of diffusion models and the sampling of diffusion models, *i.e.*, any diffusion model could use any sampling algorithm. To better demonstrate this transformation, we present the pseudocodes.

**Linearly adding noise diffusion models.** These kind of diffusion models define the forward process as a linear interpolation between the clean image and Gaussian noise, *i.e.*, $\mathbf{x}_t = \alpha_t \mathbf{x}_0 + \sigma_t \epsilon$. It could be transformed as:

$$\overbrace{\frac{\mathbf{x}_t}{\alpha_t}}^{\mathbf{x}_t \text{ in EDM}} = \mathbf{x}_0 + \overbrace{\frac{\sigma_t}{\alpha_t}}^{\sigma_t \text{ in EDM}} \epsilon.$$

Hence, we could directly pass $\frac{\mathbf{x}_t}{\alpha_t}$ and $\frac{\sigma_t}{\alpha_t}$ to an EDM model to get the predicted $\mathbf{x}_0$, as shown in Algorithm 4.

**DDPM.** DDPM define a sequence $\{\beta_t\}_{t=0}^T$ and $\mathbf{x}_t = \sqrt{\prod_{i=0}^t (1-\beta_i)}\mathbf{x}_0 + \sqrt{1 - \prod_{i=0}^t (1-\beta_i)}\epsilon$, which could be seen as a special case of Linear diffusion models.

**VP-SDE.** VP-SDE is the continuous case of DDPM, which define a stochastic differential equation (SDE) as

$$dX_t = -\frac{1}{2}\beta(t)X_t dt + \sqrt{\beta(t)}dW_t, \ t \in [0, 1],$$

where $\beta(t) = \beta_{t \cdot T} \cdot T$. Consequently, we could also use an EDM model to solve the reverse VP-SDE by predicting $\mathbf{x}_0 = \mathbf{h}_\theta\left(\frac{\mathbf{x}_t}{\sqrt{\exp\left(-\int_0^t \beta(s)ds\right)}}, \sqrt{\frac{1-\exp\left(-\int_0^t \beta(s)ds\right)}{\exp\left(-\int_0^t \beta(s)ds\right)}}\right)$.

**VE-SDE.** The forward process of VE-SDE is defined as

$$dX_t = \sqrt{\frac{d\sigma(t)^2}{dt}} dW_t.$$

EDM is a special case of VE-SDE, where $\sigma(t) = t$. For a $\mathbf{x}_t$ in VE-SDE, the variance of the noise is $\sigma(t)$. Hence, we could directly use $\mathbf{h}_\theta(\mathbf{x}_t, \sigma(t))$ to get the predicted $\mathbf{x}_0$.

# B Experimental Details

## B.1 Certified Robustness Details

We adhere to the certified robustness pipeline established by Cohen et al. [6], yet our method potentially offers a tighter certified bound, as demonstrated in Appendix A.3. To prevent confusion regarding our hyper-parameters and those in Cohen et al. [6], we clarify these hyper-parameters as follows:

$\epsilon$ = maximum allowed $\ell_2$ perturbation of the input

$N$ = number of samples used in binomial test to estimate the lower bound $\underline{p_A}$

$\sigma$ = std. of Gaussian noise data augmentation during training and certification

$T$ = number of diffusion timesteps

$T'$ = number of diffusion timesteps we selected to calculate/(estimate) evidence lower bounds

$\alpha$ = Type one error for estimating the lower bound $\underline{p_A}$.

To ensure a fair comparison, we follow previous work and do not estimate $\overline{p_B}$, instead directly setting $\overline{p_B} = 1 - \underline{p_A}$. This approach results in a considerably lower certified robustness, especially for ImageNet. Therefore, it is possible that the actual certified robustness of diffusion classifiers might be significantly higher than the results we present.

There are multiple ways to reduce the number of timesteps selected to estimate the evidence lower bounds (e.g., uniformly, selecting the first $T'$ timesteps). Chen et al. [3] demonstrate that these strategies achieve similar results. In this work, we just adopt the simplest strategy, i.e., uniformly select $T'$ timesteps.

## B.2 ImageNet Baselines

We compare our methods with training-based methods as outlined in Cohen et al. [6], Salman et al. [38], Jeong and Shin [13], Zhai et al. [50], and with diffusion-based purification methods in Carlini et al. [2], Xiao et al. [47]. Since none of these studies provide code for ImageNet64x64, we retrain ResNet-50 using these methods and certify it via randomized smoothing. For training-based methods, we utilize the implementation by Jeong and Shin [13][2] with their default hyper-parameters for ImageNet. For diffusion-based purification methods, we first resize the input images to 64x64, then resize them back to 256x256, and feed them into the subsequent classifiers. This process ensures a fair comparison with our method.

## B.3 Ablation Studies of Diffusion Models

In the ImageNet dataset, we employ the same diffusion models as our baseline studies, Carlini et al. [2] and Xiao et al. [47]. For the CIFAR-10 dataset, however, we opt for a 55M diffusion model from Karras et al. [16], as its parameterization aligns more closely with our definition of diffusion models. In contrast, Carlini et al. [2] utilizes a 50M diffusion model from Nichol and Dhariwal [33]. When we replicate the study by Carlini et al. [2] using the model from Karras et al. [16] and a WRN-70-2 [49], we achieve certified robustness of $76.56\%$, $59.76\%$, and $41.01\%$ for radii 0.25, 0.5, and 0.75, respectively, which is nearly identical to their results shown in Table 1. This finding suggests that the choice of diffusion models does not significantly impact certified robustness.

We choose the discrete version of the diffusion model from Karras et al. [16] as our definition because it is the simplest version for deriving the ELBOs in this paper. For more details, please see Appendix A.8.

---

[2]https://github.com/jh-jeong/smoothing-consistency

| Name | Properties |
|------|-----------|
| DC | ELBO: $\log p(\mathbf{x}_0|y) \geq -\sum_{t=1}^T \mathbb{E}\left[w_t\|\mathbf{h}_\theta(\mathbf{x}_t, \sigma_t, y) - \mathbf{x}_0\|_2^2\right] + C$

Diffusion Classifier: $\mathrm{DC}(\mathbf{x})_y = \frac{\exp(-\sum_{t=1}^T \mathbb{E}\left[w_t\|\mathbf{h}_\theta(\mathbf{x}_t, \sigma_t, y) - \mathbf{x}_0\|_2^2\right])}{\sum_{\hat{y}} \exp\left(-\sum_{t=1}^T \mathbb{E}\left[w_t\|\mathbf{h}_\theta(\mathbf{x}_t, \sigma_t, \hat{y}) - \mathbf{x}_0\|_2^2\right]\right)}$

Smoothed Classifier: $g_{\mathrm{DC}}(\mathbf{x}) = \mathrm{DC}(\mathbf{x})$

Certified Robustness: $R = \frac{\sqrt{2}T(\underline{p_A} - \overline{p_B})}{(2/\sqrt{D} + \sqrt{2/\pi})\sum_{i=1}^T w_i/\sigma_i}$ |
| EPNDC | ELBO: $\log p(\mathbf{x}_t|y) \geq -\sum_{i=t}^T w_i\mathbb{E}[\|\mathbb{E}[q(\mathbf{x}_i|\mathbf{x}_{i+1}, \mathbf{x}_t)] - \mathbb{E}[p(\mathbf{x}_i|\mathbf{x}_{i+1}, y)]\|^2] + C_2$

Diffusion Classifier: $\mathrm{EPNDC}(\mathbf{x}_t)_y = \frac{\exp(-\sum_{i=t}^T w_i\mathbb{E}[\|\mathbb{E}[q(\mathbf{x}_i|\mathbf{x}_{i+1}, \mathbf{x}_t)] - \mathbb{E}[p(\mathbf{x}_i|\mathbf{x}_{i+1}, y)]\|^2])}{\sum_{\hat{y}} \exp\left(-\sum_{i=t}^T w_i\mathbb{E}[\|\mathbb{E}[q(\mathbf{x}_i|\mathbf{x}_{i+1}, \mathbf{x}_t)] - \mathbb{E}[p(\mathbf{x}_i|\mathbf{x}_{i+1}, \hat{y})]\|^2]\right)}$

Smoothed Classifier: $g_{\mathrm{EPNDC}}(\mathbf{x})_y = P_{\boldsymbol{\epsilon} \sim \mathcal{N}(\mathbf{0},\mathbf{I})}\left(\arg\max_{\hat{y}} \mathrm{EPNDC}(\mathbf{x}_0 + \sigma \cdot \boldsymbol{\epsilon})_{\hat{y}} = y\right)$

Certified Robustness: $R = \frac{\sigma}{2}\left(\Phi^{-1}(\underline{p_A}) - \Phi^{-1}(\overline{p_B})\right)$ |
| APNDC | ELBO: $\mathbb{E}_{q(\hat{\mathbf{x}}_t|\mathbf{x}_0), \mathbf{x}_0 = \mathbf{h}(\mathbf{x}_t, \sigma_t)}[\log p(\mathbf{x}_t|y)] \geq -\sum_{i=t}^T \mathbb{E}\left[w_i\|\mathbf{h}_\theta(\mathbf{x}_i, \sigma_i, y) - \mathbf{x}_0\|_2^2\right] + C_3$

Diffusion Classifier: $\mathrm{APNDC}(\mathbf{x}_t)_y = \frac{\exp(-\sum_{i=t}^T \mathbb{E}\left[w_i\|\mathbf{h}_\theta(\mathbf{x}_i, \sigma_i, y) - \mathbf{h}_\theta(\mathbf{x}_t, \sigma_t)\|_2^2\right])}{\sum_{\hat{y}} \exp\left(-\sum_{i=t}^T \mathbb{E}\left[w_i\|\mathbf{h}_\theta(\mathbf{x}_i, \sigma_i, \hat{y}) - \mathbf{h}_\theta(\mathbf{x}_t, \sigma_t)\|_2^2\right]\right)}$

Smoothed Classifier: $g_{\mathrm{APNDC}}(\mathbf{x})_y = P_{\boldsymbol{\epsilon} \sim \mathcal{N}(\mathbf{0},\mathbf{I})}\left(\arg\max_{\hat{y}} \mathrm{APNDC}(\mathbf{x}_0 + \sigma \cdot \boldsymbol{\epsilon})_{\hat{y}} = y\right)$

Certified Robustness: $R = \frac{\sigma}{2}\left(\Phi^{-1}(\underline{p_A}) - \Phi^{-1}(\overline{p_B})\right)$ |

Table 4: An illustration of the relationship between different ELBOs, likelihood, classifiers, and certified robustness.

### B.4 Ablation Studies on Time Complexity Reduction Techniques

**Variance reduction proves beneficial.** As illustrated in Figure 2(a), our variance reduction technique enables a significant reduction in time complexity while maintaining high clean accuracy. Regarding certified robustness, Table 1 shows that a tenfold reduction in time complexity results in only a minor, approximately 3%, decrease in certified robustness across all radii. Notably, with $T' = 100$, our method still attains state-of-the-art clean accuracy and certified robustness for $\epsilon = 0.25, 0.5, 0.75$ while only cost one-fourth of the NFEs in Xiao et al. [47]. This underscores the effectiveness of our variance reduction approach.

**Sift-and-refine proves beneficial.** Our method's requirement for function evaluations (NFEs) is proportional to the number of classes, which limits its scalability in datasets with a large number of candidate classes. For instance, in the ImageNet dataset, without the Sift-and-refine algorithm, our method necessitates approximately $10^8$ NFEs per image (i.e., $100 \cdot 1000 \cdot 1000$), translating to about $3 \times 10^6$ seconds for certifying each image on a single 3090 GPU. In contrast, Xiao et al. [47]'s method requires about $4 \times 10^6$ NFEs (i.e., $40 \cdot 10 \cdot 10000$), or roughly $3 \times 10^5$ seconds. Our proposed Sift-and-refine technique, however, can swiftly identify the most likely candidates, thereby reducing the time complexity. It adjusts the processing time based on the difficulty of the input samples. With this technique, our method requires only about $1 \times 10^5$ seconds per image, making it more efficient compared to Xiao et al. [47].

## C  Discussions

### C.1  ELBO, Likelihood, Classifier and Certified Robustness

As demonstrated in Figure 1 and Table 4, the basic idea of all these diffusion classifiers is to approximate the log likelihood by ELBO and calculate the classification probability via Bayes' theorem. All these classifiers possess non-trivial robustness, but certified lower bounds vary in tightness.

Table 5: The accuracy of diffusion classifier [3] using the EDM checkpoint [16] on CIFAR-10 test set with various weight. Result are tested on the same subset with 512 images as in Chen et al. [3].

| Weight Name | $\hat{w}_t$ | Accuracy (%) |
|---|---|---|
| EDM | $\frac{\sigma_t^2+\sigma_d^2}{\sigma_t^2\sigma_d^2}\frac{1}{\sqrt{2\pi}k_\sigma}\exp(-\frac{\|\log\sigma_t-k_\mu\|^2}{2k_\sigma^2})$ | 94.92 |
| Uniform | $1$ | 85.76 |
| DDPM | $\frac{1}{\sigma_t}$ | 90.23 |
| EDM-W | $\frac{\sigma_t^2+\sigma_d^2}{\sigma_t^2\sigma_d^2}$ | 88.67 |
| EDM-$p(t)$ | $\frac{1}{\sqrt{2\pi}k_\sigma}\exp(-\frac{\|\log\sigma_t-k_\mu\|^2}{2k_\sigma^2})$ | 93.75 |
| ELBO | $\frac{\sigma_{t+1}-\sigma_t}{\sigma_t^3}$ | 44.53 |

The diffusion classifier is intuitively considered the most robust, as it can process not only clean images but also those corrupted by Gaussian noise. This means it can achieve high clean accuracy by leveraging less noisy samples, while also enhancing robustness through more noisy samples where adversarial perturbations are significantly masked by Gaussian noise. However, its certified robustness is not as tight as desired. As explained in Appendix A.2, this is because we assume that the maximum Lipschitz condition holds across the entire space, which is a relatively broad assumption, resulting in a less stringent certified robustness.

EPNDC and APNDC utilize the Evidence Lower Bound (ELBO) of corrupted data $\log p(\mathbf{x}_\tau)$. This implies that the least noisy examples they can process are at the noise level corresponding to $\tau$. As $\tau$ increases, the upper bound of certified robustness also rises, but this simultaneously diminishes the classifiers' ability to accurately categorize clean data. As highlighted in Chen et al. [3], this mechanism proves less effective, particularly in datasets with a large number of classes but low resolution. In such cases, the addition of even a small variance of Gaussian noise can render the entire image unclassifiable.

Generally, the Diffusion Classifier is intuitively more robust than both EPNDC and APNDC. However, obtaining a tight theoretical certified lower bound for the Diffusion Classifier proves more challenging compared to EPNDC and APNDC. Therefore, we contend that in practical applications, the Diffusion Classifier is preferable to both EPNDC and APNDC. Looking forward, we aim to derive a tighter certified lower bound for the Diffusion Classifier in future research.

## C.2 The Loss Weight in Diffusion Classifiers

Typically, when training diffusion models by Eq. (5), most researchers opt to use a re-designed weight $\hat{w}_t$ (e.g., $\hat{w}_t = 1$ in Ho et al. [11]), rather than the derived weight $w_t$. When constructing diffusion classifier from an off-the-shelf diffusion model, we find that maintaining consistency between the loss weight in diffusion classifier and the training weight is crucial. As shown in Table 5, any inconsistency in loss weight leads to a decrease in performance. Conversely, performance enhances when the diffusion classifier's weight closely aligns with the training weight. Surprisingly, the derived weight (ELBO) yields the worst performance. Therefore, when $\tau = 0$, we use the training weight $\hat{w}_t$ rather than the derived weight $w_t$.

Similarly, when $\tau \neq 0$, the derived weight $w_t^{(\tau)}$ also results in the worst performance among different weight configurations, as detailed in Appendix C.2. However, determining the optimal weight in this general case is a complex challenge. Directly replacing $w_t^{(\tau)}$ with the training weight $\hat{w}_t$ is not appropriate, as $w_t^{(\tau)} \neq w_t$. To address this issue, we propose an alternative: multiplying the weight by $\frac{\hat{w}_t}{w_t}$ (i.e., using $\frac{\hat{w}_t w_t^{(\tau)}}{w_t}$ as the weight for EPNDC). This method effectively acts as a substitution of $w_t$ with $\hat{w}_t$ when $\tau = 0$. However, it is important to note that this approach is not optimal, and deriving an optimal weight appears to be infeasible.

To enhance generative performance, most researchers opt not to use the derived weight $w_t$ for the training loss $\|\mathbb{E}[q(\mathbf{x}_t|\mathbf{x}_{t+1}, \mathbf{x}_0)] - \mathbb{E}[p(\mathbf{x}_t|\mathbf{x}_{t+1})]\|^2$. Instead, they employ a redesigned weight $\hat{w}_t$. For example, Ho et al. [11] define $\hat{w}_t = \frac{1}{\sigma_t}$. Another approach involves sampling $i$ from a specifically designed distribution $p(t)$, rather than a uniform distribution. This approach can be interpreted as

adjusting the loss weight to $\hat{w}_t = p(t)$, without altering the distribution of $i$:

$$\mathbb{E}_{\mathbf{x}, t \sim p(t)} \mathbb{E}_{\mathbf{x}_t}[\|\mathbf{h}(\mathbf{x}_t, \sigma_t) - \mathbf{x}\|_2^2]$$

$$= \mathbb{E}_{\mathbf{x}} \sum_{t=0}^{T} p(t) \mathbb{E}_{\mathbf{x}_t}[\|\mathbf{h}(\mathbf{x}_t, \sigma_t) - \mathbf{x}\|_2^2]$$

$$= \mathbb{E}_{\mathbf{x}, t} \mathbb{E}_{\mathbf{x}_t}[p(t)\|\mathbf{h}(\mathbf{x}_\tau, \sigma_t) - \mathbf{x}\|_2^2].$$

For example, Karras et al. [16] employ $\frac{\sigma_t^2 + \sigma_d^2}{\sigma_t^2 \sigma_d^2}$ as the loss weight but modify $p(t)$ to $\frac{1}{\sqrt{2\pi}k_\sigma} \exp\left(-\frac{\|\log \sigma_t - k_\mu\|^2}{2k_\sigma^2}\right)$, where $\sigma_d, k_\sigma, k_\mu$ are hyper-parameters. Thus, it is equivalent to setting $\hat{w}_t = \frac{\sigma_t^2 + \sigma_d^2}{\sigma_t^2 \sigma_d^2} \frac{1}{\sqrt{2\pi}k_\sigma} \exp\left(-\frac{\|\log \sigma_t - k_\mu\|^2}{2k_\sigma^2}\right)$.

We discover that maintaining consistency in the loss weight used in diffusion classifiers and during training is crucial. Take EDM as an example. As demonstrated in Table 5, if we fail to maintain the consistency of the loss weight between training and testing, a significant performance drop occurs. The closer the weight during testing is to the weight used during training, the better the performance.

Surprisingly, the derived weight (ELBO) yields the poorest performance, as indicated in Table 5. This issue also occurs when calculating $D_{\mathrm{KL}}(q(\mathbf{x}_t|\mathbf{x}_{t+1}, \mathbf{x}_t)\|p(\mathbf{x}_t|\mathbf{x}_{t+1}))$. The performance of our derived weight is significantly inferior to both the uniform weight and the DDPM weight. For $D_{\mathrm{KL}}(q(\mathbf{x}_t|\mathbf{x}_{t+1}, \mathbf{x}_0)\|p(\mathbf{x}_t|\mathbf{x}_{t+1}))$, substituting the derived weight $\frac{\sigma_{t+1} - \sigma_t}{\sigma_t^3}$ with the training weight $\frac{\sigma_t^2 + \sigma_d^2}{\sigma_t^2 \sigma_d^2} \frac{1}{\sqrt{2\pi}k_\sigma} \exp\left(-\frac{\|\log \sigma_t - k_\mu\|^2}{2k_\sigma^2}\right)$ is feasible. However, this substitution becomes problematic for $D_{\mathrm{KL}}(q(\mathbf{x}_t|\mathbf{x}_{t+1}, \mathbf{x}_\tau)\|p(\mathbf{x}_t|\mathbf{x}_{t+1}))$, as there is no $\frac{\sigma_{t+1} - \sigma_t}{\sigma_t^3}$ term, as outlined in Eq. (13). We propose two alternative strategies: one interprets the shift from $w_t$ to $\hat{w}_t$ as reweighting the KL divergence by $\frac{\hat{w}_t}{w_t}$, requiring only the multiplication of $\frac{\hat{w}_t}{w_t}$ to our derived weight. The other strategy involves using the equation $\hat{w}_t = w_t$ to reduce the number of parameters. While these two interpretations yield identical results for $D_{\mathrm{KL}}(q(\mathbf{x}_t|\mathbf{x}_{t+1}, \mathbf{x}_0)\|p(\mathbf{x}_t|\mathbf{x}_{t+1}))$, they differ for $D_{\mathrm{KL}}(q(\mathbf{x}_t|\mathbf{x}_{t+1}, \mathbf{x}_\tau)\|p(\mathbf{x}_t|\mathbf{x}_{t+1}))$. Therefore, directly deriving an optimal weight appears infeasible. For detailed information, see Appendix A.6.

### C.3 Certified Robustness of Chen et al. [3].

In this section, our goal is to establish a certified lower bound of Chen et al. [3]. Specifically, under the conditions where $p_A = 1$ and $\overline{p_B} = 0$, the maximum certified radius is determined to be approximately 0.39. This implies that the certified radius we derive could not surpass 0.39. In practical applications, our method achieves an average certified radius of 0.0002.

Nevertheless, we can significantly enhance the certified radius by adjusting $w_t$. For example, simply set $w_t \equiv 1$, we can get an average certified radius of 0.009, 34 times larger than previous one. By reducing $w_t$ for smaller $t$ and increasing it for larger $t$, such as zeroing the weight when $\sigma_t \leq 0.5$, we achieve an average radius of 0.156.

It is important to note that this finding does not imply that Chen et al. [3] lacks robustness in its weight-adjusted version. As discussed in previous sections, the certified radius is merely a lower bound of the actual robust radius, and a higher lower bound does not necessarily equate to a higher actual robust radius. When $w_t$ is increased for larger $t$, obtaining a tighter certified radius will be much easier, but the actual robust radius could intuitively decrease due to increased noise in the input images.

### C.4 Time Complexity Reduction Techniques that Do Not Help

**Advanced Integration.** In this work, we select $T$ timesteps uniformly. As we discuss earlier, there is a one-to-one mapping between $\sigma$ and $t$ in our context. When we change the variable to $\sigma$, this can be seen as calculating the expectation using the Euler integral. We also try using more advanced integration methods, like Gauss Quadrature, but only observe nuanced differences that there is no any difference in clean accuracy and certified robustness, indicating that the expectation calculation is robust to small truncation errors. We do not attempt any non-parallel integrals, as they significantly reduce throughput.

---

**Algorithm 5** Sift-and-refine

---

1: **Require:** An ELBO computation function for a given timestep $t$ and a class $y$, denoted as $\mathbf{e}_\theta$ (applicable for DC, EPNDC, or APNDC); a noisy input image $\mathbf{x}_\tau$; sift timesteps $\{t_i\}_{i=0}^{T_s}$; refine steps $\{t_i\}_{i=0}^{T_r}$; threshold $\tau$.
2: **Initialize:** the candidate class list $C = \{0, 1, \ldots, K\}$.
3: **for** $i = 0$ **to** $T_s$ **do**
4:     **for each** class $y$ in $C$ **do**
5:         Calculate ELBO for class $y$ at timestep $t_i$: $e_y = \mathbf{e}_\theta(\mathbf{x}_t, \sigma_{t_i}, y)$.
6:     **end for**
7:     Find the class $m$ with the minimum ELBO: $m = \arg\min_{y \in C} e_y$.
8:     Update $C$ by removing classes with a reconstruction loss $\tau$ greater than that of $m$:
       $C = \{y \in C : e_y - e_m < \tau\}$
9: **end for**
10: Reinitialize $e_y$: $e_y = \infty \ \forall y \notin C, 0 \ \forall y \in C$.
11: **for** $i = 0$ **to** $T_r$ **do**
12:     **for each** class $y$ in $C$ **do**
13:         Calculate and accumulate ELBO for class $y$ at timestep $t_i$: $e_y = e_y + \mathbf{e}_\theta(\mathbf{x}_t, t_i, y)$.
14:     **end for**
15: **end for**
16: **Return:** $\tilde{y} = \arg\min_y e_y$.

---

---

**Algorithm 6** Discrete Progressive Class Selection

---

1: **Require:** A pre-trained diffusion model $\epsilon_\theta$, input image $\mathbf{x}$, predefined number of classes $K$, class candidate trajectory $\mathcal{C}_{\text{cand}}$ and timestep candidate trajectory $\mathcal{T}_{\text{cand}}$.
2: **Initialize:** entire timesteps $\tilde{\mathbf{t}} = \text{vec}([1, 2, \cdots, T])$, counter pointer $c = 0$ and top-k cache $\mathcal{K}_{\text{top}} = \{1, 2, \cdots, K\}$.
3: **for** $(c_1, c_2)$ in $(\mathcal{C}_{\text{cand}}, \mathcal{T}_{\text{cand}})$ **do**
4:     $\tilde{\mathbf{t}}_{\text{select}} = \tilde{\mathbf{t}}[c_2 - c : c_2]$.
5:     Calculate $\text{logit}_y = \sum_{t \in \tilde{\mathbf{t}}_{\text{select}}} [w_t \|\mathbf{h}_\theta(\mathbf{x}_t, t, y) - \boldsymbol{\epsilon}\|_2^2]$ for all $y \in \mathcal{K}_{\text{top}}$ simultaneously using $\mathbf{h}_\theta$.
6:     Merge $\text{logit}_y$ calculated in the previous step into $\text{logit}_y$ currently calculated.
7:     Sort $\{\text{logit}_y\}_{y \in \mathcal{K}_{\text{top}}}$.
8:     Set the smallest $c_1$ class as new $\mathcal{K}_{\text{top}}$ from the sorted $\{\text{logit}_y\}_{y \in \mathcal{K}_{\text{top}}}$.
9:     Update counter pointer $c = c_2$.
10: **end for**
11: **Return:** $\mathcal{K}_{\text{top}}$.

---

**Sharing Noise Across Different Timesteps.** In Sec. 3.4, we demonstrate that by using the same $\mathbf{x}_i$ for all classes, we significantly reduce the variance of predictions. This allows us to use a fewer number of timesteps, thereby greatly reducing time complexity. From another perspective, using the same $\mathbf{x}_i$ for all classes is equivalent to applying the same noise to samples at the same timestep. This raises the question: "If we share the noise across all samples, could we further reduce time complexity?" However, this approach proves ineffective. Analyzing the difference between the logits of two classes reveals that sharing noise does not reduce the variance of this difference. Consequently, we opt not to share noise across different timesteps but only within the same timestep for different classes.

**Discrete progressive class selection algorithm.** We design a normalized discrete critical class selection algorithm for accelerating diffusion classifiers. Typically, the time complexity of the vanilla diffusion classifier can be defined as $\mathcal{O}(KT)$, where $K$ denotes the count of classes and $T$ denotes the number of timesteps. Sharpening $T$ is tractable, but overdoing that practice can have an extremely negative impact on final performance. Another way to speed up the computation is to actively discard some unimportant classes when estimating the conditional likelihood via conditional ELBO. Indeed, these two sub-approaches can be merged in parallel in a single algorithm (*i.e.*, our proposed class selection algorithm). Additionally, to achieve an in-depth analysis of acceleration in diffusion classifiers, the time complexity of our proposed discrete acceleration algorithm can be determined by a predefined manual class candidate trajectory $\mathcal{C}_{\text{cand}}$ as well as a predefined manual timestep candidate trajectory $\mathcal{T}_{\text{cand}}$.

The procedure of the discrete progressive class selection algorithm can be described in Algorithm 6. The specific mechanism of Algorithm 6 can be described from a simple example: the predefined trajectory $\mathcal{C}_{\text{cand}}$ is set as [400, 80, 40, 1] and the predefined trajectory $\mathcal{T}_{\text{cand}}$ is set as [2, 5, 25, 50]

Table 6: The experimental results of discrete progressive class selection algorithm.

| Timestep Trajectory | Classes Trajectory | Time Complexity | Accuracy |
|---|---|---|---|
| [2, 5, 25, 50] | [400, 80, 40, 1] | 5800 | 58.00% |
| [2, 5, 17, 50] | [500, 136, 37, 1] | 6353 | 58.20% |
| [2, 5, 17, 50] | [500, 100, 20, 1] | 5360 | 57.80% |
| [2, 4, 12, 25] | [500, 136, 37, 1] | 4569 | 54.10% |
| [2, 4, 12, 25] | [500, 100, 20, 1] | **4060** | 54.69% |
| [2, 7, 35, 50] | [297, 80, 15, 1] | 5900 | 56.84% |
| [2, 7, 35, 50] | [297, 100, 20, 1] | 6535 | 57.42% |
| [2, 6, 20, 50] | [350, 100, 20, 1] | 5400 | 57.81% |
| [2, 6, 25, 50] | [350, 100, 20, 1] | 5800 | 56.10% |
| [2, 7, 35, 50] | [297, 80, 40, 1] | 6275 | 56.84% |
| [2, 5, 25, 50] | [297, 80, 40, 1] | 5491 | 57.42% |
| [2, 5, 15, 50] | [500, 80, 40, 1] | 5700 | **59.18%** |
| [2, 4, 8, 50] | [500, 200, 100, 1] | 8000 | 56.05% |
| [2, 4, 8, 50] | [400, 160, 80, 1] | 6800 | 56.25% |
| [2, 4, 8, 50] | [400, 120, 60, 1] | 5800 | 56.05% |
| [2, 4, 8, 50] | [400, 80, 40, 1] | 4800 | 55.47% |

simultaneously. Assume we conduct this accelerated diffusion classifier with $K = 1000$ in the specific ImageNet [47]. First, we estimate the diffusion loss, which is equivalent to conditional likelihood, for 1000 classes by 2 time points and then sort the losses of the different classes to obtain the smallest 400 classes. Subsequently, we estimate the diffusion loss for 400 classes by 3 (*w.r.t.* 5-2) time points and then sort the losses for different classes to obtain the smallest 80 classes. Ultimately, the procedure concludes with the selection of 40 out of 80 classes with 20 time points (*w.r.t.* 25-5), followed by choosing 1 class from 40 classes with 25 (*w.r.t.* 50-25) time points.

The ablation outcomes presented in Table 6 illustrate that suitable design of predefined manual timestep candidate trajectories $\mathcal{C}_{cand}$ and $\mathcal{T}_{cand}$ are crucial in the final performance of diffusion classifiers. A salient observation in the initial definition of $\mathcal{C}_{cand}$ and $\mathcal{T}_{cand}$ is that $\mathcal{T}_{cand}$ can initially be small, whereas $\mathcal{C}_{cand}$ must start significantly larger to achieve high accuracy while ensuring low time complexity. Thus, the timestep trajectory [2, 5, 15, 50] and the classes trajectory [400, 80, 40, 1] are in accordance with the above conditions and achieve the best performance in various predefined trajectories.

**Accelerated Diffusion Models.** We observe that consistency models [43] and CUD [40] determine the class of the generated object at large timesteps, while they primarily refine the images at lower timesteps. In other words, at smaller timesteps, the similarity between predictions of different classes is so high that classification becomes challenging. Conversely, at larger timesteps, the images become excessively noisy, leading to less accurate predictions. Consequently, constructing generative classifiers from consistency models and CUD appears to be a difficult task.

# D  Limitations

Despite the significant improvements in certified robustness, this work still presents limitations. Firstly, the time complexity of the diffusion classifier restricts its applicability in real-world scenarios, providing primarily theoretical benefits. Additionally, the certified bounds for diffusion classifiers are not straightforward enough. Future efforts could emulate the strong law of randomized smoothing to establish a more direct certified lower bound for Chen et al. [3].

## Ethics Statements

The advancement of robust machine learning models, particularly in the realm of classification under adversarial conditions, is crucial for the safe and reliable deployment of AI in critical applications. Our work on Noised Diffusion Classifiers (NDCs) represents a significant step towards developing more secure and trustworthy AI systems. By achieving unprecedented levels of certified robustness, our approach enhances the reliability of machine learning models in adversarial environments. This is particularly beneficial in fields where decision-making reliability is paramount, such as autonomous driving, medical diagnostics, and financial fraud detection. Our methodology could substantially increase public trust in AI technologies by demonstrating resilience against adversarial attacks, thereby fostering wider acceptance and integration of AI solutions in sensitive and impactful sectors.

