# OpenReview forum: "Diffusion Models are Certifiably Robust Classifiers"
_NeurIPS.cc/2024/Conference — NeurIPS 2024 poster_

### Official Review · Reviewer_p9kw · 2024-06-27

**Soundness:** 3
**Presentation:** 3
**Contribution:** 3
**Rating:** 7
**Confidence:** 4

**Summary:**

This paper derives an upper bound of the Lipschitz constant for diffusion classifiers. Then, it proposes Exact Posterior Noised Diffusion Classifier (EPNDC) and Approximated Posterior Noised Diffusion Classifier (APNDC) by deriving ELBO upper bounds on $\log p (x_\tau)$ and thereby enabling classifying noisy images. The APNDC achieves state-of-the-art certified robustness.

**Strengths:**

The theory is cool and the math is intriguing. I like this direction because it leverages the shared Gaussian structure in diffusion models and randomized smoothing while circumventing the challenges of attacking diffusion models. The empirical evaluation results (especially Figure 2a) are also impressive.

**Weaknesses:**

In my opinion, some of the contents are not explained very clearly. Please see below and the contents of the "Questions" section.

- Table 4 is nice. However, I wish it was in the main text instead of the appendix, because the current main text misses the discussion on how to calculate the certified robustness for the proposed models.
- Figure 2 doesn't present the certified radius with a conventional diffusion classifier, as derived in Eq. (11). Since (11) is an important contribution of this work, I believe it should be included.
- I would like to see an ablation study on $\sigma_\tau$, but could not find this result.

Overall, this is still a nice paper.

**Questions:**

- Line 145: Why are the mentioned quantities bounded in the range $[0, 1]^D$? Do you clip $h_\theta (x_t, \sigma_t)$ to $[0, 1]^D$? How about $\|\| h_\theta (x_t, \sigma_t) \|\|_2^2$?
- Eq (12): The notation $q$ was introduced in Eq (1) to represent probabilities in the forward diffusion process. So, how to compute $q (x_t | x_{t+1})$? It would also be nice to add subscripts to each of the nested expectations, so that it's clear what variables the expectations are taken over.
- Also Eq (12): does the method become computationally cheaper when $\tau$ increases? Is it correct that different values of $\tau$ can reuse some computation, so if you set $\tau$ to 0, you simultaneously get the ELBO bound for $\tau = 0, 1, \ldots, T$?
- Remark 3.4: when would it make sense to use $\tau = 0$? When $\tau$ is $0$, is $\sigma_\tau$ also 0? Does this mean randomized smoothing is not used?
- Line 224: the APNDC reconstruction loss $\|\| h_\theta (x_\tau, \sigma_\tau) \|\|_2^2$ sees great resemblance to the training objective of consistency models. Does this intuitively imply that consistency models are more suitable or less suitable for APNDC? I see a short discussion about consistency models in the appendix, but it does not fully address my curiosity.
- Line 247 mentions that $\sigma_\tau \in \\{ 0.25, 0.5, 1.0 \\}$. What are the corresponding $\tau$ values? Which setting is used for which results? Are the different radii in Table 1, Table 2, and Figure 2 evaluated with the same $\sigma_\tau$ or different ones?

**Limitations:**

As is the case for numerous diffusion classifier paper, the computation complexity, while improved in this paper, is far from ideal. This paper evaluates the method on a small subset of CIFAR-10 and ImageNet, probably due to this reason.

---

> ### Author Rebuttal · Authors · 2024-08-04
>
> Thank you for recognizing the topics and contributions of our paper. We are greatly encouraged by your appreciation. Below, we address your detailed comments and hope that you find our responses satisfactory.
>
> ***Weakness 1: Table 4 should be put in the main text.***
>
> Thank you for your advice. We will move Table 4 into the main text.
>
> ***Weakness 2: Certified radii of diffusion classifiers should be included in the main table.***
>
> Thank you for your advice. We will incorporate these results into Table 1.
>
>
>
>
>
> ***Weakness 3: Ablation studies in \\(\sigma_\tau\\).***
>
> Thank you for your advice. However, we are unable to perform these experiments during the rebuttal phase because, for each \\(\sigma\_\tau\\), we would need to obtain predictions from our model over the dataset 10,000 times. Nearly all previous work uses \\(\sigma\_\tau \in \\{0.25, 0.5, 1\\}\\), and these studies have already demonstrated that these three radii are sufficient for obtaining the upper bound of the certified radius. Therefore, we did not perform this ablation study. We adopt this setting to enable fair and equitable comparisons and to avoid recoding all previous baselines.
>
>
>
>
>
>
> ***Question 1: \\(h_\theta(x_t,t)\\) are bounded.***
>
> Yes, we clip the output of the UNet \\(h_\theta(x_t,t)\\) to \\([0,1]\\), thus it is bounded.
>
> \\(\mathbb{E}\_{x\_t}[\|h\_\theta(x\_t,t)\|\_2^2]\\) is the expectation of a function whose output is bounded within \\([0,1]\\) over a Gaussian distribution, which is in the form of randomized smoothing, thus its Lipschitz constant can be bounded.
>
>
>
>
>
>
> ***Question 2: About \(q\) distribution.***
>
> Thank you for your suggestion. We will add subscripts to each of the nested expectations to clarify the expression:
>
> \\[
> \log p(\mathbf{x}\_\tau) \geq -\sum\_{t=\tau}^{T}w\_t^{(\tau)} \mathbb{E}\_{q(\mathbf{x}\_{t+1}|\mathbf{x}\_\tau)}\left[\\|\mathbb{E}\_{q(\mathbf{x}\_t|\mathbf{x}\_{t+1},\mathbf{x}\_\tau)}[\mathbf{x}\_t]-\mathbb{E}\_{p(\mathbf{x}\_{t}|\mathbf{x}\_{t+1})}[\mathbf{x}\_{t}]\\|^2\right] + C\_2,
> \\]
>
> in the revision.
>
> We do not calculate \\(q(\mathbf{x}\_t|\mathbf{x}\_{t+1})\\) directly; instead, we only calculate the expectation of the reverse Gaussian distribution \\(q(\mathbf{x}\_t|\mathbf{x}\_{t+1})\\), as shown in Eq. (13), \\(\mathbb{E}\_{q(\mathbf{x}\_t|\mathbf{x}\_{t+1},\mathbf{x}\_\tau)}[\mathbf{x}\_t]=\frac{(\sigma\_{t+1}^2-\sigma\_t^2)\mathbf{x}\_{\tau}+(\sigma\_t^2-\sigma\_\tau^2)\mathbf{x}\_{t+1}}{\sigma\_{t+1}^2-\sigma\_\tau^2}\\).
>
>
>
>
>
>
> ***Question 3: About Eq. (12).***
>
> Diffusion can be considered a continuous stochastic differential equation (SDE) within the range \\([\sigma_\tau, \sigma_T]\\). The number of function evaluations (NFEs) depends on both \\(\sigma_T - \sigma_\tau\\) and the discretization interval \\(\sigma_{i+1} - \sigma_i\\). If you keep the discretization interval unchanged, the NFEs decrease as \\(\tau\\) increases or \\(\sigma_T\\) decreases.
>
> We are impressed by your idea of reusing computations to calculate all evidence lower bounds (ELBOs) simultaneously. Since the neural network output \\(p(\mathbf{x}\_t|\mathbf{x}\_{t+1})\\) does not depend on \\(\tau\\), we can reuse this part when calculating ELBOs for different \\(\tau\\). Given that the neural network's forward pass is the computational bottleneck compared to other parts of the ELBO calculations, this means that for a given \\(\mathbf{x}\_0\\), we can compute all ELBOs over \\(\tau\\) simultaneously in the time it takes to compute a single ELBO.
>
>
>
>
>
> ***Question 4: In the case of \\(\tau=0\\).***
>
> When \\(\tau=0\\), the ELBOs of EPNDC reduce to the ELBOs in DDPM, and the EPNDC reduces to vanilla diffusion classifiers. In this case, we cannot use randomized smoothing.
>
>
>
>
>
>
> ***Question 5: Similarity between APNDC and consistency models.***
>
> We agree that there is a significant similarity between the training loss of consistency models and APNDC's ELBO. However, consistency models (CD) seem to lose their ability to classify and instead overfit to the generation task. Specifically, when \\(t\\) is small, the predictions for all labels \\(h(x_t, t, y)\\) are nearly identical, with their cosine similarity exceeding 0.99. We suspect this is due to the distillation phase causing consistency models to overfit to the generation task. We plan to investigate this issue further by training consistency models ourselves to understand the underlying reasons.
>
>
>
>
>
>
> ***Question 6: About \\(\sigma\_\tau\\).***
>
> Karras et al. [2] emphasize that since there is a bijection between \\(t\\) and \\(\sigma\\), and \\(t\\) depends on discretization, it is better to use \\(\sigma\\) as the variable to describe the noise added to the input images. Therefore, finding \\(\tau\\) for \\(\sigma\_\tau=0.25\\) is meaningless. In practice, we directly set \\(\sigma\_\tau=0.25\\) and determine \\(T'\\) discretization steps within \\([\sigma\_\tau, \sigma\_T]\\) to calculate the MSE loss at these \\(T'\\) timesteps.
>
> For Tables 1 and 2, we follow previous work by calculating the certified radius using \\(\sigma\_\tau=0.25, 0.5, 1\\), respectively, and selecting the maximum one to include in the tables and figures.
>
>
>
>
>
>
> [1] Cohen, Jeremy, Elan Rosenfeld, and Zico Kolter. "Certified adversarial robustness via randomized smoothing." international conference on machine learning. PMLR, 2019.
>
>
>
> [2] Karras, Tero, et al. "Elucidating the design space of diffusion-based generative models." Advances in neural information processing systems 35 (2022): 26565-26577.

---

### Official Review · Reviewer_76VC · 2024-07-03

**Soundness:** 3
**Presentation:** 3
**Contribution:** 3
**Rating:** 8
**Confidence:** 5

**Summary:**

The authors investigate the certified robustness of diffusion classifiers. For this purpose, they first show that these classifiers have O(1) Lipschitzness and subsequently achieve tighter robustness bounds through Bayes' theorem and the ELBO.

**Strengths:**

S1: Using diffusion models to generate large amounts of synthetic data is one of the most promising approaches to improve empirical and certified robustness in recent years. The authors utilize diffusion models directly to achieve high certified robustness.

S2: While prior work has investigated the robustness of diffusion classifiers, they do not provide certified guarantees. This gap is addressed in this work.

S3: The work provides both relevant empirical and theoretical contributions

**Weaknesses:**

W1: References could be ordered by appearance (minor)

W2: The nature of diffusion classifiers induces a considerable computational overhead compared to standard classifiers. However, the authors try to address this issue through their sift-and-refine algorithm. Still a comparison between different methods w.r.t. inference time would have been informative. (could also include standard classifiers). Note that I would not consider large computational cost as a negative point concerning paper acceptance I just believe that a comparison would be helpful for the reader. Still the appendix provides some information w.r.t. time complexity so I view this as a minor issue.

W3: Appendix D is very short and could be incorporated into the paper (at least in the camera-ready version)

**Questions:**

Q1: Could the authors provide a computational cost comparison between different methods?

**Limitations:**

Limitations are included in the appendix.

---

> ### Author Rebuttal · Authors · 2024-08-04
>
> Thank you for recognizing the contribution of our work and for providing valuable feedback. Below we address your detailed comments and hope that you find our responses satisfactory.
>
> ***Weakness 1: References could be ordered by appearance.***
>
> Thank you for your suggestion. We will revise the references to be ordered by appearance in the final version.
>
> ***Weakness 2: Appendix D can be incorporated into the main text.***
>
> Thank you for your advice. We will incorporate the Limitation section into the end of the main text in the final version.
>
>
>
> ***Weakness 3 and Q1: A table for comparing time complexity.***
>
>
>
> Thank you for your valuable suggestion. We strongly agree that a table presenting the comparison of time complexity is necessary. The result is shown below:
>
>
>
>
> CIFAR10:
>
> |              Method              | Architecture | Certifying NFEs | Certifying Real Time | Inference NFEs | Inference Real Time |
> |:---:|:---:|:--:|:---:|:----:|:----:|
> |  RS/SmoothAdv/Consistency/MACER  |  ResNet-110  |      O(N)       |        \\(10.81\\)         |      O(1)      |  \\(0.001\\)   |
> | Carlini | UNet+WRN-72  |      O(N)       |   \\(1.78 \times  10^3\\)   |      O(1)      |  \\(0.030\\)   |
> |               Xiao               | UNet+WRN-72  |     O(400N)     |   \\(7.44 \times 10^4 \\)  |     O(400)     |  \\(1.14\\)   |
> | DC/APNDC/EPNDC (variance-reduction) |     UNet     |    O(1250N)     |   \\(2.92 \times  10^4 \\)  |    O(1250)     |   \\(2.92\\)     |
>
> ImageNet:
>
> |              Method              |  Architecture  | Certifying NFEs | Certifying Real Time  | Inference NFEs | Inference Real Time |
> |:-------:|:---:|:------:|:-----:|:------:|:---------:|
> |  RS/SmoothAdv/Consistency/MACER  |   ResNet-50    |      O(N)       |   \\(26.701\\)   |      O(1)      |        \\(0.004\\)        |
> | Carlini | UNet+ResNet-50 |  O(N)  |     \\(1025.586\\)        |      O(1)      |     \\(0.108\\)  |
> | Xiao | UNet+ResNet-50 |  O(400N)   |  \\(432000\\)         |     O(400)     |    \\(43.2\\)   |
> | DC/APNDC/EPNDC (sift-and-refine) |      UNet      | undetermined  | \\(1.1 \times  10^5\\) | undetermined |   \\(112.3\\)     |
>
>
>
>
>
> As shown:
>
> - When using sift-and-refine, the NFEs for each image depend on the specific image. Thus, the time complexity is undetermined.
> - During certification, different samples are computed in parallel, so the time complexity is much lower than Inference Real Time * N.
> - The value of N for the Diffusion Classifier is ten times smaller than for other models.
> - The time complexity of diffusion classifiers is slightly higher than that of the approach by Xiao et al. (2022). We recognize this as a primary limitation of both our approach and generative classifiers in general. We are actively working to reduce the time complexity of diffusion classifiers so that it becomes independent of $K$.

---

> > ### Comment · Reviewer_76VC · 2024-08-07
> > **Concerns addressed**
> >
> > I thank the authors for their response. My concerns are appropriately addressed and I am happy to raise my score.
> > I found reading the paper quite enjoyable.

---

> > > ### Author Response · Authors · 2024-08-08
> > >
> > > Thank you for your thoughtful review and for raising your score. We are delighted to hear that you found reading our paper enjoyable and that our responses addressed your concerns. Your feedback is invaluable to us, and we appreciate your support.

---

### Official Review · Reviewer_TbjG · 2024-07-11

**Soundness:** 3
**Presentation:** 2
**Contribution:** 2
**Rating:** 6
**Confidence:** 4

**Summary:**

This work proves that diffusion classifiers possess inherent robustness to adversarial attacks by demonstrating their O(1) Lipschitzness and establishing their certified resilience. By generalizing these classifiers to handle Gaussian-corrupted data and using evidence lower bounds for likelihood approximation, the research demonstrates the superior certified robustness of Noised Diffusion Classifiers (NDCs).

**Strengths:**

The paper showcases the robustness of the proposed Noised Diffusion Classifiers (NDCs), achieving high certified robustness on the CIFAR-10 and ImageNet 64x64 datasets. The study also provides a proof of O(1) Lipschitzness for diffusion classifiers.

**Weaknesses:**

1. The proposed method combines two existing techniques, diffusion classifiers and randomized smoothing, which is not sufficiently novel. The paper needs to better highlight what sets this approach apart from existing methods and how it fundamentally advances the field. Although the authors attempt to establish a theoretical framework, the derivation of the Lipschitz constant and its implications are not sufficiently detailed, leaving unanswered questions about the robustness guarantees.

2. The experimental evaluation relies heavily on the small CIFAR-10 and ImageNet 64x64 datasets. Expanding the experiments to include larger datasets, such as ImageNet-1K, would provide a more comprehensive assessment.

3. The paper discusses techniques to reduce time complexity but does not convincingly demonstrate the practicality of the proposed methods with experimental results, such as throughput or inference latency. A more detailed analysis and comparisons of computational efficiency, especially in relation to existing methods, are needed.

**Questions:**

Please address the weaknesses mentioned above.

**Limitations:**

No potential negative societal impact.

---

> ### Author Rebuttal · Authors · 2024-08-04
>
> Thank you for appreciating the strong results of our methods. Below we address the detailed comments, and hope you may find our response satisfactory and update the score accordingly.
>
>
> ***Weakness 1: Insufficient Novelty.***
>
> We disagree with our highest respect. We justify the novelty from two aspects. First, diffusion classifiers have demonstrated superior empirical robustness and are increasingly used in robust classification tasks. However, a comprehensive theoretical understanding of their robustness is still lacking. There are concerns about whether their robustness is overestimated or if they might be vulnerable to potentially stronger adaptive attacks in the future. In this work, **we address these concerns by using proper mathematical tools (e.g., randomized smoothing) to provide a theoretical foundation for diffusion classifiers**. This contribution is novel and valuable to our community.
>
> Second, the analysis of diffusion classifiers is highly nontrivial, with several key technical contributions that are also new:
>
> 1. **Theoretical Foundation for Diffusion Classifiers:**
>    Diffusion classifiers are traditionally **not** compatible with randomized smoothing because they struggle to handle noisy data. To address this, we derive the analytical solution of the diffusion classifiers' gradient and bound their gradient norm (Lipschitz constant in Appendix A.2). This provides a lower bound and demonstrates the inherent robustness of diffusion classifiers, offering insights into the source of their robustness.
>
>
> 2. **Generalization to Noisy Data and Remarkable Certified Robustness Record:**
>
>    Bounding robustness using the maximum Lipschitz constant \\(L\\) can only provide a certified robustness of at most \\(\frac{1}{2L}\\). To achieve a tighter certified bound, we generalize diffusion classifiers to handle noisy data by deriving two noisy ELBOs, for which we can establish their point-wise Lipschitzness. These generalized diffusion classifiers achieve remarkable certified robustness, breaking previous records by exceeding 70\% certified robustness at \\(\epsilon\_2=0.5\\), which is less than 10\% below the empirical upper bound.
>
> 3. **Reduction in Time Complexity:**
>    We reduce the time complexity of diffusion classifiers by more than 10x using our proposed variance reduction and sift-and-refine methods, making diffusion classifiers more practical for large-scale applications.
>
> Overall, **we fundamentally address the key concerns about the robustness of diffusion classifiers by proving their robustness lower bound.** These contributions are significant and new, as agreed by reviewers LwbB, p9kw, and yRsf.
>
>
> ***Weakness 2: Experiments on ImageNet-256x256.***
>
> Thank you for the valuable suggestion. We agree that experiments on ImageNet-256x256 are crucial for demonstrating the scalability of diffusion classifiers. However, there are currently no conditional diffusion models available at 256x256 resolution **in the RGB space**.
>
> Existing ImageNet diffusion models are trained either at 64x64 resolution (e.g., EDM [3]) or in a 64x64 latent space (e.g., stable diffusion [2]). Although the latent diffusion models achieve state-of-the-art generation performance, they cannot be applied to adversarial defense tasks because their encoder is vulnerable to attacks [4].
>
> We recognize this as a primary limitation of our work. We will continue to address this issue by employing more advanced diffusion training methods (e.g., [1]).
>
>
>
> ***Weakness 3: Comparison of inference latency.***
>
> Thank you for the valuable suggestion. We provide further results on  time complexity:
>
> CIFAR10:
>
> |              Method              | Architecture | Certifying NFEs | Certifying Real Time | Inference NFEs | Inference Real Time |
> |:---:|:---:|:--:|:---:|:----:|:----:|
> |  RS/SmoothAdv/Consistency/MACER  |  ResNet-110  |      O(N)       |        \\(10.81\\)         |      O(1)      |  \\(0.001\\)   |
> | Carlini | UNet+WRN-72  |      O(N)       |   \\(1.78 \times  10^3\\)   |      O(1)      |  \\(0.030\\)   |
> |               Xiao               | UNet+WRN-72  |     O(400N)     |   \\(7.44 \times 10^4 \\)  |     O(400)     |  \\(1.14\\)   |
> | DC/APNDC/EPNDC (variance-reduction) |     UNet     |    O(1250N)     |   \\(2.92 \times  10^4 \\)  |    O(1250)     |   \\(2.92\\)     |
>
> ImageNet:
>
> |              Method              |  Architecture  | Certifying NFEs | Certifying Real Time  | Inference NFEs | Inference Real Time |
> |:-------:|:---:|:------:|:-----:|:------:|:---------:|
> |  RS/SmoothAdv/Consistency/MACER  |   ResNet-50    |      O(N)       |   \\(26.701\\)   |      O(1)      |        \\(0.004\\)        |
> | Carlini | UNet+ResNet-50 |  O(N)  |     \\(1025.586\\)        |      O(1)      |     \\(0.108\\)  |
> | Xiao | UNet+ResNet-50 |  O(400N)   |  \\(432000\\)         |     O(400)     |    \\(43.2\\)   |
> | DC/APNDC/EPNDC (sift-and-refine) |      UNet      | undetermined  | \\(1.1 \times  10^5\\) | undetermined |   \\(112.3\\)     |
>
>
> As shown:
>
> - When using sift-and-refine, the NFEs for each image depend on the specific image. Thus, the time complexity is undetermined.
> - During certification, different samples are computed in parallel, so the time complexity is much lower than Inference Real Time * N.
> - The value of N for the Diffusion Classifier is ten times smaller than for other models.
> - The time complexity of diffusion classifiers is slightly higher than that of the approach by Xiao et al. (2022). We recognize this as a primary limitation of both our approach and generative classifiers in general. We are actively working to reduce the time complexity so that it becomes independent of $K$.
>
>
>
> ## Reference:
>
> [1] Matryoshka diffusion models. ICLR, 2023.
>
> [2] High-resolution image synthesis with latent diffusion models. CVPR, 2022.
>
> [3] Elucidating the design space of diffusion-based generative models. NeurIPS, 2022.
>
> [4] Pixel is a Barrier: Diffusion Models Are More Adversarially Robust Than We Think. arXiv, 2024.

---

> > ### Comment · Reviewer_TbjG · 2024-08-10
> >
> > Although diffusion classifiers and randomized smoothing are two established techniques, there are indeed some gaps when applying randomized smoothing to diffusion classifiers, as mentioned in the rebuttal. Despite the lack of novelty in combining these two existing components, addressing the gaps that hinder such a combination is also a valid contribution, especially when the mitigation method is supported by a theoretical foundation.

---

> > > ### Author Response · Authors · 2024-08-10
> > >
> > > Thank you for your thoughtful review and for recognizing the value of our contributions. We are glad that you found our theoretical approach to addressing these gaps meaningful. We are also grateful for your decision to raise the score.

---

### Official Review · Reviewer_LwbB · 2024-07-13

**Soundness:** 3
**Presentation:** 3
**Contribution:** 3
**Rating:** 7
**Confidence:** 4

**Summary:**

This paper presents a theoretical analysis of the enhanced robustness in diffusion-based classifiers and introduces a generalized Noised Diffusion Classifier, EPNDC. The authors utilize the Evidence Lower Bound (ELBO) of each conditional log-likelihood $\log p(x_\tau | y) $and Bayes' theorem as the logits for each class. They identified that EPNDC is time-consuming due to the iterative computation of two conditional ELBOs. To address this, they leverage the ELBO of an ensemble of EPNDC to approximate the expected likelihood as logits without additional computational cost. Additionally, they developed variance reduction and sift-and-refine techniques to reduce time complexity. Experimental results demonstrate that APNDC achieves significantly better robustness without requiring extra training data, fewer diffusion steps, and a reduced number of samples needed to estimate the Lipschitz bound.

**Strengths:**

1. The entire paper is logically structured with a clear progression, enabling readers to understand it well. From Algorithm 1 to Algorithm 2 to Algorithm 5, the authors continuously explore problems, improve algorithms, and provide thorough analysis and theoretical proofs.

2. The experiments are comprehensive, and compared to the benchmark, EPNDC shows significant improvements in certified accuracy. This demonstrates EPNDC's high scalability in handling large datasets with numerous categories.

**Weaknesses:**

1. Some causal relationships are unclear or lack citations, requiring further explanation from the authors. For instance, in Line 156: What does the "nabla operator" refer to? It is neither explained nor cited. In Lines 161-164: "However, similar to the weak law of randomized smoothing, such certified robustness has limitations because it assumes the maximum Lipschitz condition is satisfied throughout the entire perturbation path. As a result, the robust radius is less than half of the reciprocal of the maximum Lipschitz constant." The causal relationship here is unclear and needs further clarification.

2. Although the diffusion classifier is highly scalable and robust, its clean accuracy on ImageNet is still far behind the current state-of-the-art (90%+). More details can be found at [https://paperswithcode.com/sota/image-classification-on-imagenet](https://paperswithcode.com/sota/image-classification-on-imagenet).

**Questions:**

1. Refer to weaknesses.

2. Eq (15) uses the diffusion model $h_\theta$ one more time than Eq (12). Why does APNDC not increase the computational overhead compared to EPNDC (Line201)? Please explain further.

**Limitations:**

The authors didn't address the limitations of APNDC, but Diffusion Classifiers are still far behind the current SOTA in classification accuracy. These are some potential limitations.

---

> ### Author Rebuttal · Authors · 2024-08-04
>
> Thank you for appreciating the writing and contribution of our paper. We are deeply encouraged by your kind words. Below we address detailed comments, and hope you may find our reponse satisfactory.
>
>
>
> ***Weakness 1: Nabla operator is unclear.***
>
> Thank you for pointing this out. In our paper, the Nabla operator refers to the gradient operator. We will make this clearer in the final version.
>
> ***Weakness 2: Lines 161-164 is unclear.***
>
> Thank you for your suggestion. We will modify this section in the final version to:
>
>
> > However, similar to the weak law of randomized smoothing, such certified robustness has limitations because it assumes the maximum Lipschitz condition is satisfied throughout the entire perturbation path, i.e., it assumes the equality always holds in \\(|f(\\mathbf{x}\_{adv})\_y - f(\\mathbf{x})\_y| \leq L\\|\\mathbf{x}-\\mathbf{x}\_{adv}\\|\_2\\) when \\(f\\) has Lipschitz constant \\(L\\). As a result, the equality also holds in \\( f(\\mathbf{x}\_{adv})\_y \geq f(\\mathbf{x})\_y -  L\\|\\mathbf{x}-\\mathbf{x}\_{adv}\\|\_2 \\) and \\(f(\mathbf{x}\_{adv})\_{\hat{y}} \leq f(\mathbf{x})\_{\hat{y}} +  L\\|\mathbf{x}-\mathbf{x}\_{adv}\\|\_2\\) for \\(\max\_{\hat{y} \neq y} f(\mathbf{x})\_{\hat{y}}\\). To guarantee the prediction is unchanged (i.e., \\(f(\mathbf{x}\_{adv})\_y \geq f(\mathbf{x}\_{adv})\_{\hat{y}}\\)), its requires the perturbation \\(\\|\mathbf{x}-\mathbf{x}_{adv}\\|_2\\) must be less than \\(\frac{1}{2L}\\).
>
>
> ***Weakness 3: Still not comparable with discriminative classifiers on ImageNet.***
>
>
>
> We acknowledge and agree that generative classifiers, including diffusion classifiers, currently lag behind discriminative classifiers in terms of clean accuracy on ImageNet. We recognize this as the primary limitation of our work and will include it in the limitation section of our paper. However, generative classifiers are still promising due to their robustness, interpretability, certifiability, and strong mathematical foundation, which are advantages not typically found in discriminative classifiers. We will continue to focus on advancing generative classifiers and believe they will play a crucial role in security applications.
>
>
>
>
> ***Question 1: APNDC has one more NFE than EPNDC.***
>
> Thank you for your meticulous observation. APNDC does require one more forward pass to compute \\(h\_\theta(x\_\tau, \tau)\\). We have revised our claim about the time complexity of APNDC and EPNDC to: "This nearly free ensemble can be executed with only one more forward pass of UNet."

---

> > ### Comment · Reviewer_LwbB · 2024-08-08
> >
> > Thanks for editing the article and solving my confusion. Overall, it's a great paper!

---

> > > ### Author Response · Authors · 2024-08-09
> > >
> > > Thank you for your kind words and for taking the time to review our paper. We're glad that our revisions were helpful and that you enjoyed the paper.

---

### Decision · Program_Chairs · 2024-09-25

**Decision:**

Accept (poster)

**Comment:**

This work investigates the robustness of diffusion classifiers. It is shown that diffusion classifiers are inherently robust since they possess O(1) Lipschitzness and their certified robustness is also established. Furthermore, the authors achieve tighter bounds for diffusion classifiers that classify Gaussian-corrupted data (called Noised Diffusion Classifiers; NDCs), using evidence lower bounds (ELBOs) for the appropriate distributrions and Bayes' theorem. Empirical evaluation of NDCs using CIFAR-10 shows that NDCs are certifiably robust up to 80% and 70% under $\ell_2$ perturbations that have magnitude less than 0.25 and 0.5 respectively.